# Scalable Generalized Linear Bandits:
# Online Computation and Hashing

**Kwang-Sung Jun**
UW-Madison
kjun@discovery.wisc.edu

**Aniruddha Bhargava**
UW-Madison
aniruddha@wisc.edu

**Robert Nowak**
UW-Madison
rdnowak@wisc.edu

**Rebecca Willett**
UW-Madison
willett@discovery.wisc.edu

## Abstract

Generalized Linear Bandits (GLBs), a natural extension of the stochastic linear bandits, has been popular and successful in recent years. However, existing GLBs scale poorly with the number of rounds and the number of arms, limiting their utility in practice. This paper proposes new, scalable solutions to the GLB problem in two respects. First, unlike existing GLBs, whose per-time-step space and time complexity grow at least linearly with time $t$, we propose a new algorithm that performs online computations to enjoy a constant space and time complexity. At its heart is a novel Generalized Linear extension of the Online-to-confidence-set Conversion (GLOC method) that takes *any* online learning algorithm and turns it into a GLB algorithm. As a special case, we apply GLOC to the online Newton step algorithm, which results in a low-regret GLB algorithm with much lower time and memory complexity than prior work. Second, for the case where the number $N$ of arms is very large, we propose new algorithms in which each next arm is selected via an inner product search. Such methods can be implemented via hashing algorithms (i.e., "hash-amenable") and result in a time complexity sublinear in $N$. While a Thompson sampling extension of GLOC is hash-amenable, its regret bound for $d$-dimensional arm sets scales with $d^{3/2}$, whereas GLOC's regret bound scales with $d$. Towards closing this gap, we propose a new hash-amenable algorithm whose regret bound scales with $d^{5/4}$. Finally, we propose a fast approximate hash-key computation (inner product) with a better accuracy than the state-of-the-art, which can be of independent interest. We conclude the paper with preliminary experimental results confirming the merits of our methods.

## 1 Introduction

This paper considers the problem of making generalized linear bandits (GLBs) scalable. In the stochastic GLB problem, a learner makes successive decisions to maximize her cumulative rewards. Specifically, at time $t$ the learner observes a set of arms $\mathcal{X}_t \subseteq \mathbb{R}^d$. The learner then chooses an arm $\mathbf{x}_t \in \mathcal{X}_t$ and receives a stochastic reward $y_t$ that is a noisy function of $\mathbf{x}_t$: $y_t = \mu(\mathbf{x}_t^\top \boldsymbol{\theta}^*) + \eta_t$, where $\boldsymbol{\theta}^* \in \mathbb{R}^d$ is unknown, $\mu : \mathbb{R} \to \mathbb{R}$ is a known nonlinear mapping, and $\eta_t \in \mathbb{R}$ is some zero-mean noise. This reward structure encompasses generalized linear models [29]; e.g., Bernoulli, Poisson, etc.

The key aspect of the bandit problem is that the learner does not know how much reward she would have received, had she chosen another arm. The estimation on $\boldsymbol{\theta}^*$ is thus biased by the history of the selected arms, and one needs to mix in exploratory arm selections to avoid ruling out the optimal arm. This is well-known as the exploration-exploitation dilemma. The performance of a learner is evaluated by its *regret* that measures how much cumulative reward she would have gained additionally if she had known the true $\boldsymbol{\theta}^*$. We provide backgrounds and formal definitions in Section 2.

A linear case of the problem above ($\mu(z) = z$) is called the (stochastic) linear bandit problem. Since the first formulation of the linear bandits [7], there has been a flurry of studies on the problem [11,

34, 1, 9, 5]. In an effort to generalize the restrictive linear rewards, Filippi et al. [15] propose the GLB problem and provide a low-regret algorithm, whose Thompson sampling version appears later in Abeille & Lazaric [3]. Li et al. [27] evaluates GLBs via extensive experiments where GLBs exhibit lower regrets than linear bandits for 0/1 rewards. Li et al. [28] achieves a smaller regret bound when the arm set $\mathcal{X}_t$ is finite, though with an impractical algorithm.

*However, we claim that all existing GLB algorithms [15, 28] suffer from two scalability issues that limit their practical use: (i) under a large time horizon and (ii) under a large number $N$ of arms.*

First, existing GLBs require storing all the arms and rewards appeared so far, $\{(\mathbf{x}_s, y_s)\}_{s=1}^t$, so the space complexity grows linearly with $t$. Furthermore, they have to solve a batch optimization problem for the maximum likelihood estimation (MLE) at each time step $t$ whose per-time-step time complexity grows at least linearly with $t$. While Zhang et al. [41] provide a solution whose space and time complexity do not grow over time, they consider a specific 0/1 reward with the logistic link function, and a generic solution for GLBs is not provided.

Second, existing GLBs have linear time complexities in $N$. This is impractical when $N$ is very large, which is not uncommon in applications of GLBs such as online advertisements, recommendation systems, and interactive retrieval of images or documents [26, 27, 40, 21, 25] where arms are items in a very large database. Furthermore, the interactive nature of these systems requires prompt responses as users do not want to wait. This implies that the typical linear time in $N$ is not tenable. Towards a *sublinear* time in $N$, locality sensitive hashings [18] or its extensions [35, 36, 30] are good candidates as they have been successful in fast similarity search and other machine learning problems like active learning [22], where the search time scales with $N^\rho$ for some $\rho < 1$ ($\rho$ is usually optimized and often ranges from 0.4 to 0.8 depending on the target search accuracy). Leveraging hashing in GLBs, however, relies critically on the objective function used for arm selections. The function must take a form that is readily optimized using *existing* hashing algorithms.[1] For example, algorithms whose objective function (a function of each arm $\mathbf{x} \in \mathcal{X}_t$) can be written as a distance or inner product between $\mathbf{x}$ and a query $\mathbf{q}$ are hash-amenable as there *exist* hashing methods for such functions.

To be scalable to a large time horizon, we propose a new algorithmic framework called Generalized Linear Online-to-confidence-set Conversion (GLOC) that takes in an online learning (OL) algorithm with a low 'OL' regret bound and turns it into a GLB algorithm with a low 'GLB' regret bound. The key tool is a novel generalization of the online-to-confidence-set conversion technique used in [2] (also similar to [14, 10, 16, 41]). This allows us to construct a confidence set for $\boldsymbol{\theta}^*$, which is then used to choose an arm $\mathbf{x}_t$ according to the well-known optimism in the face of uncertainty principle. By relying on an online learner, GLOC inherently performs online computations and is thus free from the scalability issues in large time steps. While any online learner equipped with a low OL regret bound can be used, we choose the online Newton step (ONS) algorithm and prove a tight OL regret bound, which results in a practical GLB algorithm with almost the same regret bound as existing inefficient GLB algorithms. We present our proposed algorithms and their regret bounds in Section 3.

For large number $N$ of arms, our proposed algorithm GLOC is not hash-amenable, to our knowledge, due to its nonlinear criterion for arm selection. As the first attempt, we derive a Thompson sampling [5, 3] extension of GLOC (GLOC-TS), which is hash-amenable due to its linear criterion. However, its regret bound scales with $d^{3/2}$ for $d$-dimensional arm sets, which is far from $d$ of GLOC. Towards closing this gap, we propose a new algorithm Quadratic GLOC (QGLOC)

| Algorithm | Regret | Hash-amenable |
|---|---|---|
| GLOC | $\tilde{O}(d\sqrt{T})$ | ✗ |
| GLOC-TS | $\tilde{O}(d^{3/2}\sqrt{T})$ | ✓ |
| QGLOC | $\tilde{O}(d^{5/4}\sqrt{T})$ | ✓ |

Table 1: Comparison of GLBs algorithms for $d$-dimensional arm sets $T$ is the time horizon. QGLOC achieves the smallest regret among hash-amenable algorithms.

with a regret bound that scales with $d^{5/4}$. We summarize the comparison of our proposed GLB algorithms in Table 1. In Section 4, we present GLOC-TS, QGLOC, and their regret bound.

Note that, while hashing achieves a time complexity sublinear in $N$, there is a nontrivial overhead of computing the projections to determine the hash keys. As an extra contribution, we reduce this overhead by proposing a new sampling-based approximate inner product method. Our proposed sampling method has smaller variance than the state-of-the-art sampling method proposed by [22, 24] when the vectors are normally distributed, which fits our setting where projection vectors are indeed normally distributed. Moreover, our method results in thinner tails in the distribution of estimation

error than the existing method, which implies a better concentration. We elaborate more on reducing the computational complexity of QOFUL in Section 5.

## 2  Preliminaries

We review relevant backgrounds here. $\mathcal{A}$ refers to a GLB algorithm, and $\mathcal{B}$ refers to an online learning algorithm. Let $\mathcal{B}_d(S)$ be the $d$-dimensional Euclidean ball of radius $S$, which overloads the notation $\mathcal{B}$. Let $\mathbf{A}_{\cdot i}$ be the $i$-th column vector of a matrix $\mathbf{A}$. Define $||\mathbf{x}||_{\mathbf{A}} := \sqrt{\mathbf{x}^\top \mathbf{A} \mathbf{x}}$ and $\mathrm{vec}(\mathbf{A}) := [\mathbf{A}_{\cdot 1}; \mathbf{A}_{\cdot 2}; \cdots ; \mathbf{A}_{\cdot d}] \in \mathbb{R}^{d^2}$ Given a function $f : \mathbb{R} \to \mathbb{R}$, we denote by $f'$ and $f''$ its first and second derivative, respectively. We define $[N] := \{1, 2, \ldots, N\}$.

**Generalized Linear Model (GLM)**  Consider modeling the reward $y$ as one-dimensional exponential family such as Bernoulli or Poisson. When the feature vector $\mathbf{x}$ is believed to correlate with $y$, one popular modeling assumption is the generalized linear model (GLM) that turns the *natural parameter* of an exponential family model into $\mathbf{x}^\top \boldsymbol{\theta}^*$ where $\boldsymbol{\theta}^*$ is a parameter [29]:

$$\mathbb{P}(y \mid z = \mathbf{x}^\top \boldsymbol{\theta}^*) = \exp\left( \frac{yz - m(z)}{g(\tau)} + h(y, \tau) \right) , \tag{1}$$

where $\tau \in \mathbb{R}^+$ is a known scale parameter and $m$, $g$, and $h$ are normalizers. It is known that $m'(z) = \mathbb{E}[y \mid z] =: \mu(z)$ and $m''(z) = \mathrm{Var}(y \mid z)$. We call $\mu(z)$ the *inverse link* function. Throughout, we assume that the exponential family being used in a GLM has a *minimal representation*, which ensures that $m(z)$ is strictly convex [38, Prop. 3.1]. Then, the negative log likelihood (NLL) $\ell(z, y) := -yz + m(z)$ of a GLM is strictly convex. We refer to such GLMs as the *canonical* GLM. In the case of Bernoulli rewards $y \in \{0, 1\}$, $m(z) = \log(1 + \exp(z))$, $\mu(z) = (1 + \exp(-z))^{-1}$, and the NLL can be written as the logistic loss: $\log(1 + \exp(-y'(\mathbf{x}_t^\top \boldsymbol{\theta}^*)))$, where $y' = 2y - 1$.

**Generalized Linear Bandits (GLB)**  Recall that $\mathbf{x}_t$ is the arm chosen at time $t$ by an algorithm. We assume that the arm set $\mathcal{X}_t$ can be of an infinite cardinality, although we focus on finite arm sets in hashing part of the paper (Section 4). One can write down the reward model (1) in a different form:

$$y_t = \mu(\mathbf{x}_t^\top \boldsymbol{\theta}^*) + \eta_t, \tag{2}$$

where $\eta_t$ is conditionally $R$-sub-Gaussian given $\mathbf{x}_t$ and $\{(\mathbf{x}_s, \eta_s)\}_{s=1}^{t-1}$. For example, Bernoulli reward model has $\eta_t$ as $1 - \mu(\mathbf{x}_t^\top \boldsymbol{\theta}^*)$ w.p. $\mu(\mathbf{x}_t^\top \boldsymbol{\theta}^*)$ and $-\mu(\mathbf{x}_t^\top \boldsymbol{\theta}^*)$ otherwise. Assume that $||\boldsymbol{\theta}^*||_2 \le S$, where $S$ is known. One can show that the sub-Gaussian scale $R$ is determined by $\mu$: $R = \sup_{z \in (-S,S)} \sqrt{\mu'(z)} \le \sqrt{L}$, where $L$ is the Lipschitz constant of $\mu$. Throughout, we assume that each arm has $\ell_2$-norm at most 1: $||\mathbf{x}||_2 \le 1, \forall \mathbf{x} \in \mathcal{X}_t, \forall t$. Let $\mathbf{x}_{t,*} := \max_{\mathbf{x} \in \mathcal{X}_t} \mathbf{x}^\top \boldsymbol{\theta}^*$. The performance of a GLB algorithm $\mathcal{A}$ is analyzed by the expected cumulative regret (or simply *regret*): $\mathrm{Regret}_T^{\mathcal{A}} := \sum_{t=1}^T \mu(\mathbf{x}_{t,*}^\top \boldsymbol{\theta}^*) - \mu((\mathbf{x}_t^{\mathcal{A}})^\top \boldsymbol{\theta}^*)$, where $\mathbf{x}_t^{\mathcal{A}}$ makes the dependence on $\mathcal{A}$ explicit.

We remark that our results in this paper hold true for a strictly larger family of distributions than the canonical GLM, which we call the *non-canonical* GLM and explain below. The condition is that the reward model follows (2) where the $R$ is now independent from $\mu$ that satisfies the following:

**Assumption 1.** *$\mu$ is L-Lipschitz on $[-S, S]$ and continuously differentiable on $(-S, S)$. Furthermore, $\inf_{z \in (-S,S)} \mu'(z) = \kappa$ for some finite $\kappa > 0$ (thus $\mu$ is strictly increasing).*

Define $\mu'(z)$ at $\pm S$ as their limits. Under Assumption 1, $m$ is defined to be an integral of $\mu$. Then, one can show that $m$ is $\kappa$-strongly convex on $\mathcal{B}_1(S)$. An example of the non-canonical GLM is the probit model for 0/1 reward where $\mu$ is the Gaussian CDF, which is popular and competitive to the Bernoulli GLM as evaluated by Li et al. [27]. Note that canonical GLMs satisfy Assumption 1.

## 3  Generalized Linear Bandits with Online Computation

We describe and analyze a new GLB algorithm called Generalized Linear Online-to-confidence-set Conversion (GLOC) that performs online computations, unlike existing GLB algorithms.

GLOC employs the optimism in the face of uncertainty principle, which dates back to [7]. That is, we maintain a confidence set $C_t$ (defined below) that traps the true parameter $\boldsymbol{\theta}^*$ with high probability (w.h.p.) and choose the arm with the largest feasible reward given $C_{t-1}$ as a constraint:

$$(\mathbf{x}_t, \tilde{\boldsymbol{\theta}}_t) := \arg \max_{\mathbf{x} \in \mathcal{X}_t, \boldsymbol{\theta} \in C_{t-1}} \langle \mathbf{x}, \boldsymbol{\theta} \rangle \tag{3}$$

The main difference between GLOC and existing GLBs is in the computation of the $C_t$'s. Prior methods involve "batch" computations that involve all past observations, and so scale poorly with

$t$. In contrast, GLOC takes in an *online* learner $\mathcal{B}$, and uses $\mathcal{B}$ as a co-routine instead of relying on a batch procedure to construct a confidence set. Specifically, at each time $t$ GLOC feeds the loss function $\ell_t(\boldsymbol{\theta}) := \ell(\mathbf{x}_t^\top \boldsymbol{\theta}, y_t)$ into the learner $\mathcal{B}$ which then outputs its parameter prediction $\boldsymbol{\theta}_t$. Let $\mathbf{X}_t \in \mathbb{R}^{t \times d}$ be the design matrix consisting of $\mathbf{x}_1, \ldots, \mathbf{x}_t$. Define $\overline{\mathbf{V}}_t := \lambda \mathbf{I} + \mathbf{X}_t^\top \mathbf{X}_t$, where $\lambda$ is the ridge parameter. Let $z_t := \mathbf{x}_t^\top \boldsymbol{\theta}_t$ and $\mathbf{z}_t := [z_1; \cdots; z_t]$. Let $\widehat{\boldsymbol{\theta}}_t := \overline{\mathbf{V}}_t^{-1} \mathbf{X}_t^\top \mathbf{z}_t$ be the ridge regression estimator taking $\mathbf{z}_t$ as responses. Theorem 1 below is the key result for constructing our confidence set $C_t$, which is a function of the parameter predictions $\{\boldsymbol{\theta}_s\}_{s=1}^t$ and the online (OL) regret bound $B_t$ of the learner $\mathcal{B}$. All the proofs are in the supplementary material (SM).

**Theorem 1.** *(Generalized Linear Online-to-Confidence-Set Conversion) Suppose we feed loss functions $\{\ell_s(\boldsymbol{\theta})\}_{s=1}^t$ into online learner $\mathcal{B}$. Let $\boldsymbol{\theta}_s$ be the parameter predicted at time step $s$ by $\mathcal{B}$. Assume that $\mathcal{B}$ has an OL regret bound $B_t$: $\forall \boldsymbol{\theta} \in \mathcal{B}_d(S), \forall t \geq 1$,*

$$\sum_{s=1}^t \ell_s(\boldsymbol{\theta}_s) - \ell_s(\boldsymbol{\theta}) \leq B_t . \tag{4}$$

*Let $\alpha(B_t) := 1 + \frac{4}{\kappa} B_t + \frac{8R^2}{\kappa^2} \log\left(\frac{2}{\delta} \sqrt{1 + \frac{2}{\kappa} B_t + \frac{4R^4}{\kappa^4 \delta^2}}\right)$. Then, with probability (w.p.) at least $1 - \delta$,*

$$\forall t \geq 1, \|\boldsymbol{\theta}^* - \widehat{\boldsymbol{\theta}}_t\|_{\overline{\mathbf{V}}_t}^2 \leq \alpha(B_t) + \lambda S^2 - \left(\|\mathbf{z}_t\|_2^2 - \widehat{\boldsymbol{\theta}}_t^\top \mathbf{X}_t^\top \mathbf{z}_t\right) =: \beta_t . \tag{5}$$

Note that the center of the ellipsoid is the ridge regression estimator on the predicted natural parameters $z_s = \mathbf{x}_s^\top \boldsymbol{\theta}_s$ rather than the rewards. Theorem 1 motivates the following confidence set:

$$C_t := \{\boldsymbol{\theta} \in \mathbb{R}^d : \|\boldsymbol{\theta} - \widehat{\boldsymbol{\theta}}_t\|_{\overline{\mathbf{V}}_t}^2 \leq \beta_t\} \tag{6}$$

which traps $\boldsymbol{\theta}^*$ for all $t \geq 1$, w.p. at least $1 - \delta$. See Algorithm 1 for pseudocode. One way to solve the optimization problem (3) is to define the function $\boldsymbol{\theta}(\mathbf{x}) := \max_{\boldsymbol{\theta} \in C_{t-1}} \mathbf{x}^\top \boldsymbol{\theta}$, and then use the Lagrangian method to write:

$$\mathbf{x}_t^{\text{GLOC}} := \arg\max_{\mathbf{x} \in \mathcal{X}_t} \mathbf{x}^\top \widehat{\boldsymbol{\theta}}_{t-1} + \sqrt{\beta_{t-1}} \|\mathbf{x}\|_{\overline{\mathbf{V}}_{t-1}^{-1}} . \tag{7}$$

We prove the regret bound of GLOC in the following theorem.

**Theorem 2.** *Let $\{\overline{\beta}_t\}$ be a nondecreasing sequence such that $\overline{\beta}_t \geq \beta_t$. Then, w.p. at least $1 - \delta$,*

$$\text{Regret}_T^{\text{GLOC}} = O\left(L\sqrt{\overline{\beta}_T dT \log T}\right)$$

Although any low-regret online learner can be combined with GLOC, one would like to ensure that $\overline{\beta}_T$ is $O(\text{polylog}(T))$ in which case the total regret can be bounded by $\tilde{O}(\sqrt{T})$. This means that we must use online learners whose OL regret grows logarithmically in $T$ such as [20, 31]. In this work, we consider the online Newton step (ONS) algorithm [20].

**Online Newton Step (ONS) for Generalized Linear Models** Note that ONS requires the loss functions to be $\alpha$-exp-concave. One can show that $\ell_t(\boldsymbol{\theta})$ is $\alpha$-exp-concave [20, Sec. 2.2]. Then, GLOC can use ONS and its OL regret bound to solve the GLB problem. However, motivated by the fact that the OL regret bound $B_t$ appears in the radius $\sqrt{\beta_t}$ of the confidence set while a tighter confidence set tends to reduce the bandit regret in practice, we derive a tight data-dependent OL regret bound tailored to GLMs.

We present our version of ONS for GLMs (ONS-GLM) in Algorithm 2. $\ell'(z, y)$ is the first derivative w.r.t. $z$ and the parameter $\epsilon$ is for inverting matrices conveniently (usually $\epsilon = 1$ or 0.1). The only difference from the original ONS [20] is that we rely on the strong convexity of $m(z)$ instead of the $\alpha$-exp-concavity of the loss thanks to the GLM structure.[2] Theorem 3 states that we achieve the desired polylogarithmic regret in $T$.

---

**Algorithm 1** GLOC

1: **Input**: $R > 0$, $\delta \in (0, 1)$, $S > 0$, $\lambda > 0$, $\kappa > 0$, an online learner $\mathcal{B}$ with known regret bounds $\{B_t\}_{t \geq 1}$.
2: Set $\overline{\mathbf{V}}_0 = \lambda \mathbf{I}$.
3: **for** $t = 1, 2, \ldots$ **do**
4:     Compute $\mathbf{x}_t$ by solving (3).
5:     Pull $\mathbf{x}_t$ and then observe $y_t$.
6:     Receive $\boldsymbol{\theta}_t$ from $\mathcal{B}$.
7:     Feed into $\mathcal{B}$ the loss $\ell_t(\boldsymbol{\theta}) = \ell(\mathbf{x}_t^\top \boldsymbol{\theta}, y_t)$.
8:     Update $\overline{\mathbf{V}}_t = \overline{\mathbf{V}}_{t-1} + \mathbf{x}_t \mathbf{x}_t^\top$ and $z_t = \mathbf{x}_t^\top \boldsymbol{\theta}_t$
9:     Compute $\widehat{\boldsymbol{\theta}}_t = \overline{\mathbf{V}}_t^{-1} \mathbf{X}_t^\top \mathbf{z}_t$ and $\beta_t$ as in (5).
10:    Define $C_t$ as in (6).
11: **end for**

---

**Algorithm 2** ONS-GLM

1: **Input**: $\kappa > 0$, $\epsilon > 0$, $S > 0$.
2: $\mathbf{A}_0 = \epsilon \mathbf{I}$.
3: Set $\boldsymbol{\theta}_1 \in \mathcal{B}_d(S)$ arbitrarily.
4: **for** $t = 1, 2, 3, \ldots$ **do**
5:     Output $\boldsymbol{\theta}_t$.
6:     Observe $\mathbf{x}_t$ and $y_t$.
7:     Incur loss $\ell(\mathbf{x}_t^\top \boldsymbol{\theta}_t, y_t)$ .
8:     $\mathbf{A}_t = \mathbf{A}_{t-1} + \mathbf{x}_t \mathbf{x}_t^\top$
9:     $\boldsymbol{\theta}'_{t+1} = \boldsymbol{\theta}_t - \frac{\ell'(\mathbf{x}_t^\top \boldsymbol{\theta}_t, y_t)}{\kappa} \mathbf{A}_t^{-1} \mathbf{x}_t$
10:    $\boldsymbol{\theta}_{t+1} = \arg\min_{\boldsymbol{\theta} \in \mathcal{B}_d(S)} \|\boldsymbol{\theta} - \boldsymbol{\theta}'_{t+1}\|_{\mathbf{A}_t}^2$
11: **end for**

**Theorem 3.** *Define* $g_s := \ell'(\mathbf{x}_s^\top \boldsymbol{\theta}_s, y_s)$. *The regret of ONS-GLM satisfies, for any* $\epsilon > 0$ *and* $t \geq 1$,

$$\sum_{s=1}^t \ell_s(\boldsymbol{\theta}_s) - \ell_s(\boldsymbol{\theta}^*) \leq \frac{1}{2\kappa} \sum_{s=1}^t g_s^2 ||\mathbf{x}_s||_{\mathbf{A}_s^{-1}}^2 + 2\kappa S^2 \epsilon =: B_t^{\text{ONS}} ,$$

*where* $B_t^{\text{ONS}} = O(\frac{L^2 + R^2 \log(t)}{\kappa} d \log t), \forall t \geq 1$ *w.h.p. If* $\max_{s \geq 1} |\eta_s|$ *is bounded by* $\bar{R}$ *w.p. 1,* $B_t^{\text{ONS}} = O(\frac{L^2 + \bar{R}^2}{\kappa} d \log t)$.

We emphasize that the OL regret bound is data-dependent. A confidence set constructed by combining Theorem 1 and Theorem 3 directly implies the following regret bound of GLOC with ONS-GLM.

**Corollary 1.** *Define* $\beta_t^{\text{ONS}}$ *by replacing* $B_t$ *with* $B_t^{\text{ONS}}$ *in (5). With probability at least* $1 - 2\delta$,

$$\forall t \geq 1, \boldsymbol{\theta}^* \in C_t^{\text{ONS}} := \left\{ \boldsymbol{\theta} \in \mathbb{R}^d : ||\boldsymbol{\theta} - \widehat{\boldsymbol{\theta}}_t||_{\bar{\mathbf{V}}_t}^2 \leq \beta_t^{\text{ONS}} \right\} . \tag{8}$$

**Corollary 2.** *Run GLOC with* $C_t^{\text{ONS}}$. *Then, w.p. at least* $1 - 2\delta$, $\forall T \geq 1$, $\text{Regret}_T^{\text{GLOC}} = \hat{O}\left( \frac{L(L+R)}{\kappa} d\sqrt{T} \log^{3/2}(T) \right)$ *where* $\hat{O}$ *ignores* $\log\log(t)$. *If* $|\eta_t|$ *is bounded by* $\bar{R}$, $\text{Regret}_T^{\text{GLOC}} = \hat{O}\left( \frac{L(L+\bar{R})}{\kappa} d\sqrt{T} \log(T) \right)$.

We make regret bound comparisons ignoring $\log\log T$ factors. For generic arm sets, our dependence on $d$ is optimal for linear rewards [34]. For the Bernoulli GLM, our regret has the same order as Zhang et al. [41]. One can show that the regret of Filippi et al. [15] has the same order as ours if we use their assumption that the reward $y_t$ is bounded by $R_{\max}$. For unbounded noise, Li et al. [28] have regret $O((LR/\kappa)d\sqrt{T}\log T)$, which is $\sqrt{\log T}$ factor smaller than ours and has $LR$ in place of $L(L+R)$. While $L(L+R)$ could be an artifact of our analysis, the gap is not too large for canonical GLMs. Let $L$ be the smallest Lipschitz constant of $\mu$. Then, $R = \sqrt{L}$. If $L \leq 1$, $R$ satisfies $R > L$, and so $L(L+R) = O(LR)$. If $L > 1$, then $L(L+R) = O(L^2)$, which is larger than $LR = O(L^{3/2})$. For the Gaussian GLM with known variance $\sigma^2$, $L = R = 1$.[3] For finite arm sets, SupCB-GLM of Li et al. [28] achieves regret of $\tilde{O}(\sqrt{dT\log N})$ that has a better scaling with $d$ but is not a practical algorithm as it wastes a large number of arm pulls. Finally, we remark that none of the existing GLB algorithms are scalable to large $T$. Zhang et al. [41] is scalable to large $T$, but is restricted to the Bernoulli GLM; e.g., theirs does not allow the probit model (non-canonical GLM) that is popular and shown to be competitive to the Bernoulli GLM [27].

**Discussion** The trick of obtaining a confidence set from an online learner appeared first in [13, 14] for the linear model, and then was used in [10, 16, 41]. GLOC is slightly different from these studies and rather close to Abbasi-Yadkori et al. [2] in that the confidence set is a function of a known regret bound. This generality frees us from re-deriving a confidence set for every online learner. Our result is essentially a nontrivial extension of Abbasi-Yadkori et al. [2] to GLMs.

One might have notice that $C_t$ does not use $\boldsymbol{\theta}_{t+1}$ that is available before pulling $\mathbf{x}_{t+1}$ and has the most up-to-date information. This is inherent to GLOC as it relies on the OL regret bound directly. One can modify the proof of ONS-GLM to have a tighter confidence set $C_t$ that uses $\boldsymbol{\theta}_{t+1}$ as we show in SM Section E. However, this is now specific to ONS-GLM, which looses generality.

## 4 Hash-Amenable Generalized Linear Bandits

We now turn to a setting where the arm set is finite but very large. For example, imagine an interactive retrieval scenario [33, 25, 6] where a user is shown $K$ images (e.g., shoes) at a time and provides relevance feedback (e.g., yes/no or 5-star rating) on each image, which is repeated until the user is satisfied. In this paper, we focus on showing one image (i.e., arm) at a time.[4] Most existing algorithms require maximizing an objective function (e.g., (7)), the complexity of which scales linearly with the number $N$ of arms. This can easily become prohibitive for large numbers of images. Furthermore, the system has to perform real-time computations to promptly choose which image to show the user in the next round. Thus, it is critical for a practical system to have a time complexity sublinear in $N$.

One naive approach is to select a subset of arms ahead of time, such as volumetric spanners [19]. However, this is specialized for an efficient exploration only and can rule out a large number of good arms. Another option is to use hashing methods. Locality-sensitive hashing and Maximum

Inner Product Search (MIPS) are effective and well-understood tools but can only be used when the objective function is a distance or an inner product computation; (7) cannot be written in this form. In this section, we consider alternatives to GLOC which are compatible with hashing.

**Thompson Sampling**    We present a Thompson sampling (TS) version of GLOC called GLOC-TS that chooses an arm $\mathbf{x}_t = \arg\max_{\mathbf{x}\in\mathcal{X}_t} \mathbf{x}^\top \dot{\boldsymbol{\theta}}_t$ where $\dot{\boldsymbol{\theta}}_t \sim \mathcal{N}(\widehat{\boldsymbol{\theta}}_{t-1}, \beta_{t-1}\overline{\mathbf{V}}_{t-1}^{-1})$. TS is known to perform well in practice [8] and can solve the polytope arm set case in polynomial time[5] whereas algorithms that solve an objective function like (3) (e.g., [1]) cannot since they have to solve an NP-hard problem [5]. We present the regret bound of GLOC-TS below. Due to space constraints, we present the pseudocode and the full version of the result in SM.

**Theorem 4.** *(Informal) If we run GLOC-TS with* $\dot{\boldsymbol{\theta}}_t \sim \mathcal{N}(\widehat{\boldsymbol{\theta}}_{t-1}, \beta_{t-1}^{\mathrm{ONS}}\overline{\mathbf{V}}_{t-1}^{-1})$, $\mathrm{Regret}_T^{\mathrm{GLOC\text{-}TS}} = \hat{O}\left(\frac{L(L+R)}{\kappa}d^{3/2}\sqrt{T}\log^{3/2}(T)\right)$ *w.h.p. If* $\eta_t$ *is bounded by* $\bar{R}$, *then* $\hat{O}\left(\frac{L(L+\bar{R})}{\kappa}d^{3/2}\sqrt{T}\log(T)\right)$.

Notice that the regret now scales with $d^{3/2}$ as expected from the analysis of linear TS [4], which is higher than scaling with $d$ of GLOC. This is concerning in the interactive retrieval or product recommendation scenario since the relevance of the shown items is harmed, which makes us wonder if one can improve the regret without loosing the hash-amenability.

**Quadratic GLOC**    We now propose a new hash-amenable algorithm called Quadratic GLOC (QGLOC). Recall that GLOC chooses the arm $\mathbf{x}^{\mathrm{GLOC}}$ by (7). Define $r = \min_{\mathbf{x}\in\mathcal{X}} ||\mathbf{x}||_2$ and

$$\overline{m}_{t-1} := \min_{\mathbf{x}:||\mathbf{x}||_2\in[r,1]} ||\mathbf{x}||_{\overline{\mathbf{V}}_{t-1}^{-1}} , \tag{9}$$

which is $r$ times the square root of the smallest eigenvalue of $\overline{\mathbf{V}}_{t-1}^{-1}$. It is easy to see that $\overline{m}_{t-1} \leq ||\mathbf{x}||_{\overline{\mathbf{V}}_{t-1}^{-1}}$ for all $\mathbf{x} \in \mathcal{X}$ and that $\overline{m}_{t-1} \geq r/\sqrt{t+\lambda}$ using the definition of $\overline{\mathbf{V}}_{t-1}$. There is an alternative way to define $\overline{m}_{t-1}$ without relying on $r$, which we present in SM.

Let $c_0 > 0$ be the exploration-exploitation tradeoff parameter (elaborated upon later). At time $t$, QGLOC chooses the arm

$$\mathbf{x}_t^{\mathrm{QGLOC}} := \arg\max_{\mathbf{x}\in\mathcal{X}_t} \langle\widehat{\boldsymbol{\theta}}_{t-1}, \mathbf{x}\rangle + \frac{\beta_{t-1}^{1/4}}{4c_0\overline{m}_{t-1}}||\mathbf{x}||_{\overline{\mathbf{V}}_{t-1}^{-1}}^2 = \arg\max_{\mathbf{x}\in\mathcal{X}_t} \langle\mathbf{q}_t, \phi(\mathbf{x})\rangle , \tag{10}$$

where $\mathbf{q}_t = [\widehat{\boldsymbol{\theta}}_{t-1}; \mathrm{vec}(\frac{\beta_{t-1}^{1/4}}{4c_0\overline{m}_{t-1}}\overline{\mathbf{V}}_{t-1}^{-1})] \in \mathbb{R}^{d+d^2}$ and $\phi(\mathbf{x}) := [\mathbf{x}; \mathrm{vec}(\mathbf{x}\mathbf{x}^\top)]$. The key property of QGLOC is that the objective function is now quadratic in $\mathbf{x}$, thus the name *Quadratic* GLOC, and can be written as an inner product. Thus, QGLOC is hash-amenable. We present the regret bound of QGLOC (10) in Theorem 5. The key step of the proof is that the QGLOC objective function (10) plus $c_0\beta^{3/4}\overline{m}_{t-1}$ is a tight upper bound of the GLOC objective function (7).

**Theorem 5.** *Run QGLOC with* $C_t^{\mathrm{ONS}}$. *Then, w.p. at least* $1-2\delta$, $\mathrm{Regret}_T^{QGLOC} = O\left(\left(\left(\frac{1}{c_0}\left(\frac{L+R}{\kappa}\right)^{1/2} + c_0\left(\frac{L+R}{\kappa}\right)^{3/2}\right)Ld^{5/4}\sqrt{T}\log^2(T)\right)\right)$. *By setting* $c_0 = \left(\frac{L+R}{\kappa}\right)^{-1/2}$, *the regret bound is* $O(\frac{L(L+R)}{\kappa}d^{5/4}\sqrt{T}\log^2(T))$.

Note that one can have a better dependence on $\log T$ when $\eta_t$ is bounded (available in the proof). The regret bound of QGLOC is a $d^{1/4}$ factor improvement over that of GLOC-TS; see Table 1. Furthermore, in (10) $c_0$ is a free parameter that adjusts the balance between the exploitation (the first term) and exploration (the second term). Interestingly, the regret guarantee *does not break down* when adjusting $c_0$ in Theorem 5. Such a characteristic is not found in existing algorithms but is attractive to practitioners, which we elaborate in SM.

**Maximum Inner Product Search (MIPS) Hashing**    While MIPS hashing algorithms such as [35, 36, 30] can solve (10) in time sublinear in $N$, these necessarily introduce an approximation error. Ideally, one would like the following guarantee on the error with probability at least $1 - \delta_{\mathrm{H}}$:

**Definition 1.** *Let* $\mathcal{X} \subseteq \mathbb{R}^{d'}$ *satisfy* $|\mathcal{X}| < \infty$. *A data point* $\tilde{\mathbf{x}} \in \mathcal{X}$ *is called* $c_{\mathrm{H}}$-*MIPS w.r.t. a given query* $\mathbf{q}$ *if it satisfies* $\langle\mathbf{q}, \tilde{\mathbf{x}}\rangle \geq c_{\mathrm{H}} \cdot \max_{\mathbf{x}\in\mathcal{X}} \langle\mathbf{q}, \mathbf{x}\rangle$ *for some* $c_{\mathrm{H}} < 1$. *An algorithm is called* $c_{\mathrm{H}}$-*MIPS if, given a query* $\mathbf{q} \in \mathbb{R}^{d'}$, *it retrieves* $\mathbf{x} \in \mathcal{X}$ *that is* $c_{\mathrm{H}}$-*MIPS w.r.t.* $\mathbf{q}$.

Unfortunately, existing MIPS algorithms do not directly offer such a guarantee, and one must build a series of hashing schemes with varying hashing parameters like Har-Peled et al. [18]. Under the fixed budget setting $T$, we elaborate our construction that is simpler than [18] in SM.

**Time and Space Complexity**    Our construction involves saving Gaussian projection vectors that are used for determining hash keys and saving the buckets containing pointers to the actual arm vectors. The time complexity for retrieving a $c_\text{H}$-MIPS solution involves determining hash keys and evaluating inner products with the arms in the retrieved buckets. Let $\rho^* < 1$ be an optimized value for the hashing (see [35] for detail). The time complexity for $d'$-dimensional vectors is $O\left(\log\left(\frac{\log(dT)}{\log(c_\text{H}^{-1})}\right) N^{\rho^*} \log(N) d'\right)$, and the space complexity (except the original data) is $O\left(\frac{\log(dT)}{\log(c_\text{H}^{-1})} N^{\rho^*} (N + \log(N) d')\right)$. While the time and space complexity grows with the time horizon $T$, the dependence is mild; $\log\log(T)$ and $\log(T)$, respectively. QGLOC uses $d' = d + d^2$,[6] and GLOC-TS uses $d' = d'$. While both achieve a time complexity sublinear in $N$, the time complexity of GLOC-TS scales with $d$ that is better than scaling with $d^2$ of QGLOC. However, GLOC-TS has a $d^{1/4}$-factor worse regret bound than QGLOC.

**Discussion**    While it is reasonable to incur small errors in solving the arm selection criteria like (10) and sacrifice some regret in practice, the regret bounds of QGLOC and GLOC-TS do not hold anymore. Though not the focus of our paper, we prove a regret bound under the presence of the hashing error in the fixed budget setting for QGLOC; see SM. Although the result therein has an inefficient space complexity that is linear in $T$, it provides the first low regret bound with time sublinear in $N$, to our knowledge.

## 5   Approximate Inner Product Computations with L1 Sampling

While hashing allows a time complexity sublinear in $N$, it performs an additional computation for determining the hash keys. Consider a hashing with $U$ tables and length-$k$ hash keys. Given a query $\mathbf{q}$ and projection vectors $\mathbf{a}^{(1)}, \ldots, \mathbf{a}^{(Uk)}$, the hashing computes $\mathbf{q}^\top \mathbf{a}^{(i)}$, $\forall i \in [Uk]$ to determine the hash key of $\mathbf{q}$. To reduce such an overhead, approximate inner product methods like [22, 24] are attractive since hash keys are determined by discretizing the inner products; small inner product errors often do not alter the hash keys.

Figure 1: (a) A box plot of estimators. L1 and L2 have the same variance, but L2 has thicker tails. (b) The frequency of L1 inducing smaller variance than L2 in 1000 trials. After 100 dimensions, L1 mostly has smaller variance than L2.

In this section, we propose an improved approximate inner product method called *L1 sampling* which we claim is more accurate than the sampling proposed by Jain et al. [22], which we call *L2 sampling*. Consider an inner product $\mathbf{q}^\top \mathbf{a}$. The main idea is to construct an unbiased estimate of $\mathbf{q}^\top \mathbf{a}$. That is, let $\mathbf{p} \in \mathbb{R}^d$ be a probability vector. Let

$$i_k \overset{\text{i.i.d.}}{\sim} \text{Multinomial}(\mathbf{p}) \quad \text{and} \quad G_k := q_{i_k} a_{i_k} / p_{i_k}, \ k \in [m] \ . \tag{11}$$

It is easy to see that $\mathbb{E} G_k = \mathbf{q}^\top \mathbf{a}$. By taking $\frac{1}{m} \sum_{k=1}^m G_k$ as an estimate of $\mathbf{q}^\top \mathbf{a}$, the time complexity is now $O(mUk)$ rather than $O(d'Uk)$. The key is to choose the right $\mathbf{p}$. L2 sampling uses $\mathbf{p}^{(\text{L2})} := [q_i^2 / ||\mathbf{q}||_2^2]_i$. Departing from L2, we propose $\mathbf{p}^{(\text{L1})}$ that we call L1 sampling and define as follows:

$$\mathbf{p}^{(\text{L1})} := [|q_1|; \cdots; |q_{d'}|] / ||\mathbf{q}||_1 \ . \tag{12}$$

We compare L1 with L2 in two different point of view. Due to space constraints, we summarize the key ideas and defer the details to SM.

The first is on their concentration of measure. Lemma 1 below shows an error bound of L1 whose failure probability decays exponentially in $m$. This is in contrast to decaying polynomially of L2 [22], which is inferior.[7]

**Lemma 1.** *Define $G_k$ as in (11) with $\mathbf{p} = \mathbf{p}^{(\text{L1})}$. Then, given a target error $\epsilon > 0$,*

$$\mathbb{P}\left(\left|\frac{1}{m} \sum_{k=1}^m G_k - \mathbf{q}^\top \mathbf{a}\right| \geq \epsilon\right) \leq 2 \exp\left(-\frac{m\epsilon^2}{2||\mathbf{q}||_1^2 ||\mathbf{a}||_{\max}^2}\right) \tag{13}$$

To illustrate such a difference, we fix $\mathbf{q}$ and $\mathbf{a}$ in 1000 dimension and apply L2 and L1 sampling 20K times each with $m = 5$ where we scale down the L2 distribution so its variance matches that of L1.

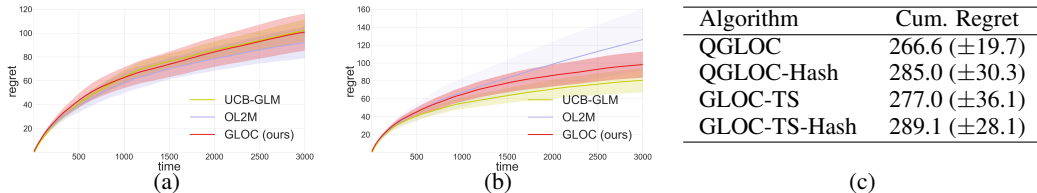

| Algorithm | Cum. Regret |
|---|---|
| QGLOC | 266.6 ($\pm$19.7) |
| QGLOC-Hash | 285.0 ($\pm$30.3) |
| GLOC-TS | 277.0 ($\pm$36.1) |
| GLOC-TS-Hash | 289.1 ($\pm$28.1) |

(a)    (b)    (c)

Figure 2: Cumulative regrets with confidence intervals under the (a) logit and (b) probit model. (c) Cumulative regrets with confidence intervals of hash-amenable algorithms.

Figure 1(a) shows that L2 has thicker tails than L1. Note this is not a pathological case but a typical case for Gaussian $\mathbf{q}$ and $\mathbf{a}$. This confirms our claim that L1 is safer than L2.

Another point of comparison is the variance of L2 and L1. We show that the variance of L1 may or may not be larger than L2 in SM; there is no absolute winner. However, if $\mathbf{q}$ and $\mathbf{a}$ follow a Gaussian distribution, then L1 induces smaller variances than L2 for large enough $d$; see Lemma 9 in SM. Figure 1(b) confirms such a result. The actual gap between the variance of L2 and L1 is also nontrivial under the Gaussian assumption. For instance, with $d = 200$, the average variance of $G_k$ induced by L2 is 0.99 whereas that induced by L1 is 0.63 on average. Although a stochastic assumption on the vectors being inner-producted is often unrealistic, in our work we deal with projection vectors $\mathbf{a}$ that are truly normally distributed.

# 6   Experiments

We now show our experiment results comparing GLB algorithms and hash-amenable algorithms.

**GLB Algorithms**   We compare GLOC with two different algorithms: UCB-GLM [28] and Online Learning for Logit Model (OL2M) [41].[8] For each trial, we draw $\boldsymbol{\theta}^* \in \mathbb{R}^d$ and $N$ arms ($\mathcal{X}$) uniformly at random from the unit sphere. We set $d = 10$ and $\mathcal{X}_t = \mathcal{X}, \forall t \geq 1$. Note it is a common practice to scale the confidence set radius for bandits [8, 27]. Following Zhang et al. [41], for OL2M we set the squared radius $\gamma_t = c \log(\det(\mathbf{Z}_t)/\det(\mathbf{Z}_1))$, where $c$ is a tuning parameter. For UCB-GLM, we set the radius as $\alpha = \sqrt{cd \log t}$. For GLOC, we replace $\beta_t^{\text{ONS}}$ with $c \sum_{s=1}^{t} g_s^2 ||\mathbf{x}_s||_{\mathbf{A}_s^{-1}}^2$. While parameter tuning in practice is nontrivial, for the sake of comparison we tune $c \in \{10^1, 10^{0.5}, \ldots, 10^{-3}\}$ and report the best one. We perform 40 trials up to time $T = 3000$ for each method and compute confidence bounds on the regret.

We consider two GLM rewards: $(i)$ the logit model (the Bernoulli GLM) and $(ii)$ the probit model (non-canonical GLM) for 0/1 rewards that sets $\mu$ as the probit function. Since OL2M is for the logit model only, we expect to see the consequences of model mismatch in the probit setting. For GLOC and UCB-GLM, we specify the correct reward model. We plot the cumulative regret under the logit model in Figure 2(a). All three methods perform similarly, and we do not find any statistically significant difference based on paired t test. The result for the probit model in Figure 2(b) shows that OL2M indeed has higher regret than both GLOC and UCB-GLM due to the model mismatch in the probit setting. Specifically, we verify that at $t = 3000$ the difference between the regret of UCB-GLM and OL2M is statistically significant. Furthermore, OL2M exhibits a significantly higher variance in the regret, which is unattractive in practice. This shows the importance of being generalizable to *any* GLM reward. Note we observe a big increase in running time for UCB-GLM compared to OL2M and GLOC.

**Hash-Amenable GLBs**   To compare hash-amenable GLBs, we use the logit model as above but now with $N$=100,000 and $T$=5000. We run QGLOC, QGLOC with hashing (QGLOC-Hash), GLOC-TS, and GLOC-TS with hashing (GLOC-TS-Hash), where we use the hashing to compute the objective function (e.g., (10)) on just 1% of the data points and save a significant amount of computation. Details on our hashing implementation is found in SM. Figure 2(c) summarizes the result. We observe that QGLOC-Hash and GLOC-TS-Hash increase regret from QGLOC and GLOC-TS, respectively, but only moderately, which shows the efficacy of hashing.

# 7   Future Work

In this paper, we have proposed scalable algorithms for the GLB problem: $(i)$ for large time horizon $T$ and $(ii)$ for large number $N$ of arms. There exists a number of interesting future work. First,

we would like to extend the GLM rewards to the single index models [23] so one does not need to know the function $\mu$ ahead of time under mild assumptions. Second, closing the regret bound gap between QGLOC and GLOC without loosing hash-amenability would be interesting: i.e., develop a hash-amenable GLB algorithm with $O(d\sqrt{T})$ regret. In this direction, a first attempt could be to design a hashing scheme that can directly solve (7) approximately.

**Acknowledgments**   This work was partially supported by the NSF grant IIS-1447449 and the MURI grant 2015-05174-04. The authors thank Yasin Abbasi-Yadkori and Anshumali Shrivastava for providing constructive feedback and Xin Hunt for her contribution at the initial stage.

## Footnotes

[1] Without this designation, no *currently known* bandit algorithm achieves a sublinear time complexity in $N$.

[2] A similar change to ONS has been applied in [16, 41].

[3] The reason why $R$ is not $\sigma$ here is that the sufficient statistic of the GLM is $y/\sigma$, which is equivalent to dealing with the normalized reward. Then, $\sigma$ appears as a factor in the regret bound.

[4] One image at a time is a simplification of the practical setting. One can extend it to showing multiple images at a time, which is a special case of the combinatorial bandits of Qin et al. [32].

[5]ConfidenceBall$_1$ algorithm of Dani et al. [11] can solve the problem in polynomial time as well.

[6] Note that this does not mean we need to store $\text{vec}(\mathbf{x}\mathbf{x}^\top)$ since an inner product with it is structured.

[7] In fact, one can show a bound for L2 that fails with exponentially-decaying probability. However, the bound introduces a constant that can be arbitrarily large, which makes the tails thick. We provide details on this in SM.

[8]We have chosen UCB-GLM over GLM-UCB of Filippi et al. [15] as UCB-GLM has a lower regret bound.

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
