[Supplementary Material]

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

# Supplementary Material (Appendix)

## A   Proof of Theorem 1

We first describe the sketch of the proof. We perform a generalized version of the online-to-confidence-set conversion of Abbasi-Yadkori et al. [2, Theorem 1] that was strictly for linear rewards. Unlike that work, which deals with the squared loss, we now work with the negative log likelihood loss. The key is to use the strong-convexity of the loss function to turn the OL regret bound (4) into a quadratic equation in $\boldsymbol{\theta}^*$. Then, it remains to bound random quantities with a Martingale concentration bound.

*Proof.* Let $\ell'(z, y)$ and $\ell''(z, y)$ be the first and second derivative of the loss $\ell$ w.r.t. $z$. We lower bound the LHS of (4) using Taylor's theorem with some $\xi_s$ between $\mathbf{x}_s^\top \boldsymbol{\theta}^*$ and $\mathbf{x}_s^\top \boldsymbol{\theta}_s$:

$$B_t \geq \sum_{s=1}^{t} \ell_s(\boldsymbol{\theta}_s) - \ell_s(\boldsymbol{\theta}^*) = \sum_{s=1}^{t} \ell'(\mathbf{x}_s^\top \boldsymbol{\theta}^*, y_s) \mathbf{x}_s^\top (\boldsymbol{\theta}_s - \boldsymbol{\theta}^*) + \frac{\ell''(\xi_s, y_s)}{2} (\mathbf{x}_s^\top (\boldsymbol{\theta}_s - \boldsymbol{\theta}^*))^2$$

$$\overset{\text{(Assumption 1)}}{\geq} \sum_{s=1}^{t} \ell'(\mathbf{x}_s^\top \boldsymbol{\theta}^*, y_s) \mathbf{x}_s^\top (\boldsymbol{\theta}_s - \boldsymbol{\theta}^*) + \frac{\kappa}{2} (\mathbf{x}_s^\top (\boldsymbol{\theta}_s - \boldsymbol{\theta}^*))^2 .$$

Since $\ell'(\mathbf{x}_s^\top \boldsymbol{\theta}^*, y_s) = -y_s + \mu(\mathbf{x}_s^\top \boldsymbol{\theta}^*) = -\eta_s$,

$$\sum_{s=1}^{t} (\mathbf{x}_s^\top (\boldsymbol{\theta}_s - \boldsymbol{\theta}^*))^2 \leq \frac{2}{\kappa} B_t + \frac{2}{\kappa} \sum_{s=1}^{t} \eta_s \left( \mathbf{x}_s^\top (\boldsymbol{\theta}_s - \boldsymbol{\theta}^*) \right) .$$

Note that the second term in the RHS involves $\eta_t$ that is unknown and random, which we bound using Abbasi-Yadkori et al. [2, Corollary 8]. That is, w.p. at least $1 - \delta$, for all $t \geq 1$,

$$\sum_{s=1}^{t} \eta_s \left( \mathbf{x}_s^\top (\boldsymbol{\theta}_s - \boldsymbol{\theta}^*) \right) \leq R \sqrt{\left( 2 + 2 \sum_{s=1}^{t} (\mathbf{x}_s^\top (\boldsymbol{\theta}_s - \boldsymbol{\theta}^*))^2 \right) \cdot \log \left( \frac{1}{\delta} \sqrt{1 + \sum_{s=1}^{t} (\mathbf{x}_s^\top (\boldsymbol{\theta}_s - \boldsymbol{\theta}^*))^2} \right)} .$$

Then,

$$\sum_{s=1}^{t} \left( \mathbf{x}_s^\top (\boldsymbol{\theta}_s - \boldsymbol{\theta}^*) \right)^2 \leq \frac{2}{\kappa} B_t + \frac{2R}{\kappa} \sqrt{\left( 2 + 2 \sum_{s=1}^{t} (\mathbf{x}_s^\top (\boldsymbol{\theta}_s - \boldsymbol{\theta}^*))^2 \right) \cdot \log \left( \frac{1}{\delta} \sqrt{1 + \sum_{s=1}^{t} (\mathbf{x}_s^\top (\boldsymbol{\theta}_s - \boldsymbol{\theta}^*))^2} \right)} .$$

Define $q := \sqrt{1 + \sum_{s=1}^{t} (\mathbf{x}_s^\top (\boldsymbol{\theta}_s - \boldsymbol{\theta}^*))^2}$. Then, the inequality above can be written as $q^2 \leq 1 + \frac{2}{\kappa} B_t + \frac{2\sqrt{2}R}{\kappa} q \sqrt{\log(q/\delta)}$. The following Lemma is useful. See Section A.1 for a proof.

**Lemma 2.** *Let $\delta \in (0, 1), a \geq 0, f \geq 0, q \geq 1$. Then,*

$$q^2 \leq a + fq \sqrt{\log \left( \frac{q}{\delta} \right)} \implies q^2 \leq 2a + f^2 \log \left( \frac{\sqrt{4a + f^4/(4\delta^2)}}{\delta} \right)$$

Applying Lemma 2 with $a := 1 + \frac{2}{\kappa} B_t$ and $f := 2\sqrt{2}R/\kappa$, we have

$$\sum_{s=1}^{t} (\mathbf{x}_s^\top (\boldsymbol{\theta}_s - \boldsymbol{\theta}^*))^2 \leq 1 + \frac{4}{\kappa} B_t + \frac{8R^2}{\kappa^2} \log \left( \frac{1}{\delta} \sqrt{4 + \frac{8}{\kappa} B_t + \frac{64R^4}{\kappa^4 \cdot 4\delta^2}} \right) = \beta_t'$$

Then, one can rewrite the above as

$$||\mathbf{z}_t - \mathbf{X}_t \boldsymbol{\theta}^*||_2^2 \leq \beta_t' .$$

Let $\lambda > 0$. We add $\lambda ||\boldsymbol{\theta}^*||$ to the both sides.

$$\lambda ||\boldsymbol{\theta}^*||_2^2 + ||\mathbf{z}_t - \mathbf{X}_t \boldsymbol{\theta}^*||_2^2 \leq \lambda ||\boldsymbol{\theta}^*||_2^2 + \beta_t' \leq \lambda S^2 + \beta_t' .$$

Hereafter, we omit $t$ from $\mathbf{X}_t$ and $\mathbf{z}_t$ for brevity. Since the LHS is quadratic in $\boldsymbol{\theta}^*$, we can rewrite it as an ellipsoid centered at $\widehat{\boldsymbol{\theta}}_t := \arg\min_{\boldsymbol{\theta}} \lambda ||\boldsymbol{\theta}||_2^2 + ||\mathbf{z} - \mathbf{X}\boldsymbol{\theta}||_2^2 = \overline{\mathbf{V}}_t^{-1} \mathbf{X}^\top \mathbf{z}$ as follows:

$$||\boldsymbol{\theta}^* - \widehat{\boldsymbol{\theta}}_t||_{\overline{\mathbf{V}}_t}^2 + \underbrace{\lambda ||\widehat{\boldsymbol{\theta}}_t||_2^2 + ||\mathbf{z} - \mathbf{X}\widehat{\boldsymbol{\theta}}_t||_2^2}_{= ||\mathbf{z}||_2^2 - \widehat{\boldsymbol{\theta}}_t^\top \mathbf{X}^\top \mathbf{z}} \leq \lambda S^2 + \beta_t' ,$$

which concludes the proof. $\qquad\square$

## A.1 Proof of Lemma 2

*Proof.* Let $c := f\sqrt{\log(q/\delta)}$. Then, $q^2 \le a + cq \implies q^2 - cq - a \le 0$. Solving it for $q$, we get $q \le \frac{c+\sqrt{c^2+4a}}{2}$. Then, using $(u+v)^2 \le 2(u^2 + v^2)$,

$$q^2 \le \left(\frac{c + \sqrt{c^2 + 4a}}{2}\right)^2 \le \frac{2(c^2 + c^2 + 4a)}{4} = c^2 + 2a \tag{14}$$

$$\iff q^2 \le 2a + f^2\log(q/\delta)$$

One might suspect that $c$ has $q$ in it, which might cause a problem. To be assured, one can prove the contrapositive: $q > \frac{c+\sqrt{c^2+4a}}{2} \implies q^2 - cq - a > 0$. To see this, $q^2 - cq - a = (q - \frac{c+\sqrt{c^2+4a}}{2})(q - \frac{c-\sqrt{c^2+4a}}{2})$ and since $q > \frac{c+\sqrt{c^2+4a}}{2}$ it suffices to show that $q - \frac{c-\sqrt{c^2+4a}}{2} > 0$. Then, since $q - \frac{c-\sqrt{c^2+4a}}{2} \ge q - \frac{c+\sqrt{c^2+4a}}{2} > 0$.

Using $\log u \le \frac{1}{2}u$,

$$q^2 \le 2a + f^2\log(q/\delta) \le 2a + \frac{f^2}{2\delta}q \iff q^2 - \frac{f^2}{2\delta}q - 2a \le 0 \ .$$

Solving the quadratic inequality for $q$, we have $q \le \frac{f^2/(2\delta)+\sqrt{(f^4/(4\delta^2))+8a}}{2}$. This implies that

$$q^2 \le \frac{2(f^4/(4\delta^2) + f^4/(4\delta^2) + 8a)}{4} = \frac{f^4}{4\delta^2} + 4a \ .$$

Now, applying this inequality on $q$ in the RHS of (14),

$$q^2 \le 2a + f^2\log\left(\frac{\sqrt{4a + f^4/(4\delta^2)}}{\delta}\right)$$

$\square$

## B  Proof of Theorem 2

*Proof.* Our proof closely follow a standard technique (cf. Abbasi-Yadkori et al. [1]). Define $\mathbf{x}_{t,*} = \arg\max_{\mathbf{x}\in\mathcal{X}_t} \langle \mathbf{x}, \boldsymbol{\theta}^* \rangle$. Let $r_t := \mu(\mathbf{x}_{t,*}^\top\boldsymbol{\theta}^*) - \mu(\mathbf{x}_t^\top\boldsymbol{\theta}^*)$ be the instantaneous regret. Using $\mu(\mathbf{x}_{t,*}^\top\boldsymbol{\theta}^*) - \mu(\mathbf{x}_t^\top\boldsymbol{\theta}^*) \le L(\mathbf{x}_{t,*}^\top\boldsymbol{\theta}^* - \mathbf{x}_t^\top\boldsymbol{\theta}^*)$,

$$\begin{aligned}
\frac{r_t}{L} &\le \mathbf{x}_{t,*}^\top\boldsymbol{\theta}^* - \mathbf{x}_t^\top\boldsymbol{\theta}^* \\
&\le \mathbf{x}_t^\top\tilde{\boldsymbol{\theta}}_t - \mathbf{x}_t^\top\boldsymbol{\theta}^* \\
&= \mathbf{x}_t^\top(\tilde{\boldsymbol{\theta}}_t - \widehat{\boldsymbol{\theta}}_{t-1}) + \mathbf{x}_t^\top(\widehat{\boldsymbol{\theta}}_{t-1} - \boldsymbol{\theta}^*) \\
&\le \|\mathbf{x}_t\|_{\overline{\mathbf{V}}_{t-1}^{-1}}\|\tilde{\boldsymbol{\theta}}_t - \widehat{\boldsymbol{\theta}}_{t-1}\|_{\overline{\mathbf{V}}_{t-1}} + \|\mathbf{x}_t\|_{\overline{\mathbf{V}}_{t-1}^{-1}}\|\boldsymbol{\theta}^* - \widehat{\boldsymbol{\theta}}_{t-1}\|_{\overline{\mathbf{V}}_{t-1}} \\
&\le 2\sqrt{\bar{\beta}_t}\|\mathbf{x}_t\|_{\overline{\mathbf{V}}_{t-1}^{-1}} \ .
\end{aligned}$$

Note that

$$\sum_{t=1}^{T}\log\left(1 + \|\mathbf{x}_t\|_{\overline{\mathbf{V}}_{t-1}^{-1}}^2\right) = \log\left(\frac{\det(\overline{\mathbf{V}}_T)}{\det(\lambda\mathbf{I})}\right) \le d\log\left(1 + T/(d\lambda)\right) \ , \tag{15}$$

which is due to Abbasi-Yadkori et al. [1, Lemma 11].

The following lemmas become useful.

**Lemma 3.** *For any $q, x \ge 0$,*

$$\min\{q, x\} \le \max\{2, q\}\log(1 + x)$$

*Proof.* It is not hard to see that

$$x \in [0, a] \implies x \le \frac{a}{\log(1+a)}\log(1+x) \tag{16}$$

We consider the following two cases.

**Case 1.** $q \leq 2$

If $x \leq 2$, by (16), $\min\{2, x\} = x \leq \frac{2}{\log(3)} \log(1 + x) \leq 2\log(1 + x)$. If $x > 2$, $\min\{2, x\} = 2 \leq 2\log(1 + 2) \leq 2\log(1 + x)$. Thus, for any $x$, $\min\{q, x\} \leq \min\{2, x\} \leq 2\log(1 + x)$.

**Case 2.** $q > 2$

If $x \leq q$, by (16), $\min\{q, x\} = x \leq \frac{q}{\log(1+q)} \log(1 + x) < q\log(1 + x)$. If $x > q$, $\min\{q, x\} = q \leq q\log(1 + 2) \leq q\log(1 + x)$.

Combining both cases to complete the proof. $\qquad\square$

**Lemma 4.** *If A is a value independent of t,*

$$\sum_{t=1}^{T} \min\{A, ||\mathbf{x}_t||^2_{\overline{\mathbf{V}}_{t-1}^{-1}}\} \leq \max\{2, A\} d \log(1 + T/(d\lambda)) .$$

*Proof.* Combine Lemma 3 and (15). $\qquad\square$

Since $r_t$ cannot be bigger than $2LS$,

$$\sum_{t=1}^{T} r_t \leq \sum_{t=1}^{T} \min\{2LS, 2L\sqrt{\overline{\beta}_t}||\mathbf{x}_t||_{\overline{\mathbf{V}}_t^{-1}}\}$$

$$\leq 2L\sqrt{\overline{\beta}_T} \sum_{t=1}^{T} \min\{S/\sqrt{\overline{\beta}_T}, ||\mathbf{x}_t||_{\overline{\mathbf{V}}_t^{-1}}\}$$

$$\overset{\text{(C.-S.)}}{\leq} 2L\sqrt{\overline{\beta}_T} \sqrt{T \sum_{t=1}^{T} \min\left\{\frac{S^2}{\overline{\beta}_T}, ||\mathbf{x}_t||^2_{\overline{\mathbf{V}}_t^{-1}}\right\}}$$

$$\overset{\text{(Lem. 4)}}{\leq} 2L\sqrt{\overline{\beta}_T} \sqrt{T \max\{2, S^2/\overline{\beta}_T\} d \log(1 + T/(d\lambda))}$$

$$= O\left(L\sqrt{\overline{\beta}_T}\sqrt{T} \cdot \sqrt{d\log T}\right)$$

where C.-S. stands for the Cauchy-Schwartz inequality. $\qquad\square$

## C Proof of Theorem 3

*Proof.* We closely follow the proof of Hazan et al. [20]. Since $\ell(z, y)$ is $\kappa$-strongly convex w.r.t. $z \in \mathcal{B}_1(S)$,

$$\ell(\mathbf{x}_s^\top \boldsymbol{\theta}_s, y_s) - \ell(\mathbf{x}_s^\top \boldsymbol{\theta}^*, y_s) \leq \ell'(\mathbf{x}_s^\top \boldsymbol{\theta}_s, y_s) \cdot \mathbf{x}_s^\top (\boldsymbol{\theta}_s - \boldsymbol{\theta}^*) - \frac{\kappa}{2}(\mathbf{x}_s^\top (\boldsymbol{\theta}_s - \boldsymbol{\theta}^*))^2 . \quad (17)$$

Define $g_s := \ell'(\mathbf{x}_s^\top \boldsymbol{\theta}_s, y_s)$. Note that by the update rule of Algorithm 2,

$$\boldsymbol{\theta}'_{s+1} - \boldsymbol{\theta}^* = \boldsymbol{\theta}_s - \boldsymbol{\theta}^* - \frac{g_s}{\kappa}\mathbf{A}_s^{-1}\mathbf{x}_s$$

$$\implies ||\boldsymbol{\theta}'_{s+1} - \boldsymbol{\theta}^*||^2_{\mathbf{A}_s} = ||\boldsymbol{\theta}_s - \boldsymbol{\theta}^*||^2_{\mathbf{A}_s} - \frac{2g_s}{\kappa}\mathbf{x}_s^\top (\boldsymbol{\theta}_s - \boldsymbol{\theta}^*) + \frac{g_s^2}{\kappa^2}||\mathbf{x}_s||^2_{\mathbf{A}_s^{-1}} \quad (18)$$

By the property of the generalized projection (see Hazan et al. [20, Lemma 8])

$$||\boldsymbol{\theta}'_{s+1} - \boldsymbol{\theta}^*||^2_{\mathbf{A}_s} \geq ||\boldsymbol{\theta}_{s+1} - \boldsymbol{\theta}^*||^2_{\mathbf{A}_s} .$$

Now, together with (18),

$$||\boldsymbol{\theta}_{s+1} - \boldsymbol{\theta}^*||^2_{\mathbf{A}_s} \leq ||\boldsymbol{\theta}_s - \boldsymbol{\theta}^*||^2_{\mathbf{A}_s} - \frac{2g_s}{\kappa}\mathbf{x}_s^\top (\boldsymbol{\theta}_s - \boldsymbol{\theta}^*) + \frac{g_s^2}{\kappa^2}||\mathbf{x}_s||^2_{\mathbf{A}_s^{-1}}$$

$$\implies \sum_{s=1}^{t} g_s\mathbf{x}_s^\top (\boldsymbol{\theta}_s - \boldsymbol{\theta}^*) \leq \sum_{s=1}^{t} \frac{g_s^2}{2\kappa}||\mathbf{x}_s||^2_{\mathbf{A}_s^{-1}} + \underbrace{\frac{\kappa}{2}\sum_{s=1}^{t} ||\boldsymbol{\theta}_s - \boldsymbol{\theta}^*||^2_{\mathbf{A}_s} - ||\boldsymbol{\theta}_{s+1} - \boldsymbol{\theta}^*||^2_{\mathbf{A}_s}}_{=:D_1} .$$

Note

$$D_1 = ||\boldsymbol{\theta}_1 - \boldsymbol{\theta}^*||_{\mathbf{A}_1}^2 + \left( \sum_{s=2}^{t} ||\boldsymbol{\theta}_s - \boldsymbol{\theta}^*||_{\mathbf{A}_s}^2 - \sum_{s=2}^{t} ||\boldsymbol{\theta}_s - \boldsymbol{\theta}^*||_{\mathbf{A}_{s-1}}^2 \right) - ||\boldsymbol{\theta}_{t+1} - \boldsymbol{\theta}^*||_{\mathbf{A}_t}^2$$

$$\leq ||\boldsymbol{\theta}_1 - \boldsymbol{\theta}^*||_{\mathbf{A}_1}^2 + \left( \sum_{s=2}^{t} ||\boldsymbol{\theta}_s - \boldsymbol{\theta}^*||_{\mathbf{A}_s}^2 - \sum_{s=2}^{t} ||\boldsymbol{\theta}_s - \boldsymbol{\theta}^*||_{\mathbf{A}_{s-1}}^2 \right)$$

$$\overset{(a)}{=} ||\boldsymbol{\theta}_1 - \boldsymbol{\theta}^*||_{\mathbf{A}_1}^2 + \left( -||\boldsymbol{\theta}_1 - \boldsymbol{\theta}^*||_{\mathbf{x}_1\mathbf{x}_1^\top}^2 + \sum_{s=1}^{t} ||\boldsymbol{\theta}_s - \boldsymbol{\theta}^*||_{\mathbf{x}_s\mathbf{x}_s^\top}^2 \right)$$

$$= ||\boldsymbol{\theta}_1 - \boldsymbol{\theta}^*||_{\epsilon\mathbf{I}}^2 + \sum_{s=1}^{t} ||\boldsymbol{\theta}_s - \boldsymbol{\theta}^*||_{\mathbf{x}_s\mathbf{x}_s^\top}^2 \leq 4\epsilon S^2 + \sum_{s=1}^{t} ||\boldsymbol{\theta}_s - \boldsymbol{\theta}^*||_{\mathbf{x}_s\mathbf{x}_s^\top}^2$$

where $(a)$ is due to $\mathbf{A}_s - \mathbf{A}_{s-1} = \mathbf{x}_s\mathbf{x}_s^\top$. Therefore,

$$\sum_{s=1}^{t} g_s \mathbf{x}_s^\top (\boldsymbol{\theta}_s - \boldsymbol{\theta}^*) \leq \sum_{s=1}^{t} \frac{g_s^2}{2\kappa} ||\mathbf{x}_s||_{\mathbf{A}_s^{-1}}^2 + 2\epsilon\kappa S^2 + \frac{\kappa}{2} \sum_{s=1}^{t} ||\boldsymbol{\theta}_s - \boldsymbol{\theta}^*||_{\mathbf{x}_s\mathbf{x}_s^\top}^2 \ .$$

Move the rightmost sum in the RHS to the LHS to see that the LHS now coincide with the RHS of (17). This leads to

$$\sum_{s=1}^{t} \ell(\mathbf{x}_s^\top \boldsymbol{\theta}_s, y_s) - \ell(\mathbf{x}_s^\top \boldsymbol{\theta}^*, y_s) \leq \frac{1}{2\kappa} \sum_{s=1}^{t} g_s^2 ||\mathbf{x}_s||_{\mathbf{A}_s^{-1}}^2 + 2\epsilon\kappa S^2 = B_t^{\mathrm{ONS}} \ .$$

This yields the statement of the theorem.

For characterizing the order of $B_t^{\mathrm{ONS}}$, notice that $g_s$ is a random variable:

$$g_s^2 = (-y_s + \mu(\mathbf{x}_s^\top \boldsymbol{\theta}_s))^2 = (-\mu(\mathbf{x}_s^\top \boldsymbol{\theta}^*) - \eta_s + \mu(\mathbf{x}_s^\top \boldsymbol{\theta}_s))^2$$

$$\leq 2(\mu(\mathbf{x}_s^\top \boldsymbol{\theta}_s) - \mu(\mathbf{x}_s^\top \boldsymbol{\theta}^*))^2 + 2\eta_s^2$$

$$\leq 2(L \cdot \mathbf{x}_s^\top (\boldsymbol{\theta}_s - \boldsymbol{\theta}^*))^2 + 2\eta_s^2$$

$$\leq 2L^2 \cdot 4S^2 + 2\eta_s^2$$

Let $\delta < 1$ be the target failure rate. By the sub-Gaussianity of $\eta_s$,

$$\mathbb{P}\left( \forall s \geq 1, |\eta_s|^2 \geq 2R^2 \log(4s^2/\delta) \right) \leq \sum_{s \geq 1} \mathbb{P}\left( |\eta_s|^2 \geq 2R^2 \log(4s^2/\delta) \right) \leq \sum_{s \geq 1} \delta/(2s^2) \leq \delta \ .$$

Thus, w.p. at least $1 - \delta$, $\max_{s \leq t} g_s^2 \leq 8L^2 S^2 + 4R^2 \log(4t^2/\delta) = O(L^2 + R^2 \log(t/\delta))$. Furthermore, $\sum_{s=1}^{t} ||\mathbf{x}_s||_{\mathbf{A}_s^{-1}}^2 \leq d\log(1 + (t/\epsilon))$ by Hazan et al. [20, Lemma 11]. Thus, w.p. at least $1 - \delta$, $\forall t \geq 1, B_t^{\mathrm{ONS}} = O\left( \frac{L^2 + R^2 \log(t/\delta)}{\kappa} d\log t \right)$.

For the case where $|\eta_s|$ is bounded by $\bar{R}$ w.p. 1 (e.g., $\bar{R} = \frac{1}{2}$ for Bernoulli), $\max_{s \leq t} g_s^2 \leq 8L^2 S^2 + 2\bar{R}^2$, which leads to $B_t^{\mathrm{ONS}} = O\left( \frac{L^2 + \bar{R}^2}{\kappa} d\log t \right)$. $\qquad\square$

## D   Proof of Corollaries 1 and 2

The proof of Corollary 1 a trivial consequence of combining Theorem 1 and Theorem 3.

Corollary 2 is simply a combination of Theorem 2 and Corollary 1. Note that $\overline{\beta}_t^{\mathrm{ONS}} = \alpha(B_t^{\mathrm{ONS}}) + \lambda S^2 \geq \beta_t^{\mathrm{ONS}}$ by noticing that $||\mathbf{z}_t||_2^2 - \widehat{\boldsymbol{\theta}}_t^\top \mathbf{X}_t^\top \mathbf{z}_t$ is nonnegative (from the proof of Theorem 2). This concludes the proof.

## E   A Tighter Confidence Set

While the confidence set constructed by Theorem 2 is generic and allows us to rely on any online learner with a known regret bound, one can find a tighter confidence set by analyzing the online learner directly. We show one instance of such for ONS. A distinctive characteristic of our new confidence set, denoted by $C_t^{\mathrm{ONS}+}$, is that it now depends on $y_t$ (note that $C_t^{\mathrm{ONS}}$ depends on $y_1, \ldots, y_{t-1}$ only).

We deviate from the proof of Theorem 3. Recall that

$$D_1 = ||\boldsymbol{\theta}_1 - \boldsymbol{\theta}^*||^2_{\mathbf{A}_1} + \left(\sum_{s=2}^{t}||\boldsymbol{\theta}_s - \boldsymbol{\theta}^*||^2_{\mathbf{A}_s} - \sum_{s=2}^{t}||\boldsymbol{\theta}_s - \boldsymbol{\theta}^*||^2_{\mathbf{A}_{s-1}}\right) - ||\boldsymbol{\theta}_{t+1} - \boldsymbol{\theta}^*||^2_{\mathbf{A}_t}$$

We previously dropped the term $||\boldsymbol{\theta}_{t+1} - \boldsymbol{\theta}^*||^2_{\mathbf{A}_t}$. We we now keep it, which leads to:

$$D_1 \le 4\epsilon S^2 + \sum_{s=1}^{t}||\boldsymbol{\theta}_s - \boldsymbol{\theta}^*||^2_{\mathbf{x}_s\mathbf{x}_s^\top} - ||\boldsymbol{\theta}_{t+1} - \boldsymbol{\theta}^*||^2_{\mathbf{A}_t}$$

Following the same argument,

$$\left(\sum_{s=1}^{t}\ell_s(\mathbf{x}_s^\top\boldsymbol{\theta}_t) - \ell_s(\mathbf{x}_s^\top\boldsymbol{\theta}^*)\right) + \frac{\kappa}{2}||\boldsymbol{\theta}_{t+1} - \boldsymbol{\theta}^*||^2_{\mathbf{A}_t} \le \sum_{s=1}^{t}\frac{g_s^2}{2\kappa}||\mathbf{x}_s||^2_{\mathbf{A}_s^{-1}} + 2\epsilon\kappa S^2 = B_t^{\text{ONS}}$$

$$\implies \left(\sum_{s=1}^{t}\ell_s(\mathbf{x}_s^\top\boldsymbol{\theta}_t) - \ell_s(\mathbf{x}_s^\top\boldsymbol{\theta}^*)\right) \le B_t^{\text{ONS}} - \frac{\kappa}{2}||\boldsymbol{\theta}_{t+1} - \boldsymbol{\theta}^*||^2_{\mathbf{A}_t} \ ,$$

Combining the above with the proof of Theorem 1,

$$\sum_{s=1}^{t}(\mathbf{x}_s^\top(\boldsymbol{\theta}_s - \boldsymbol{\theta}^*))^2$$

$$\le 1 + \frac{4}{\kappa}(B_t^{\text{ONS}} - \frac{\kappa}{2}||\boldsymbol{\theta}_{t+1} - \boldsymbol{\theta}^*||^2_{\mathbf{A}_t}) + \frac{8R^2}{\kappa^2}\log\left(\frac{2}{\delta}\sqrt{1 + \frac{2}{\kappa}B_t^{\text{ONS}} + \frac{4R^4}{\kappa^4\delta^2}}\right)$$

$$\iff \sum_{s=1}^{t}(\mathbf{x}_s^\top(\boldsymbol{\theta}_s - \boldsymbol{\theta}^*))^2 + 2||\boldsymbol{\theta}_{t+1} - \boldsymbol{\theta}^*||^2_{\mathbf{A}_t} \le 1 + \frac{4}{\kappa}B_t^{\text{ONS}} + \frac{8R^2}{\kappa^2}\log\left(\frac{2}{\delta}\sqrt{1 + \frac{2}{\kappa}B_t^{\text{ONS}} + \frac{4R^4}{\kappa^4\delta^2}}\right)$$

Define $\mathbf{z}_t = [\mathbf{x}_s^\top\boldsymbol{\theta}_s]_{s\in[t]}$, $\mathbf{z}_t^* = [\mathbf{x}_s^\top\boldsymbol{\theta}^*]_{s\in[t]}$ and $\mathbf{z}_s' = [\mathbf{x}_s^\top\boldsymbol{\theta}_{t+1}]_{s\in[t]}$. Then, the LHS above is

$$||\mathbf{z}_t - \mathbf{z}_t^*||_2^2 + 2||\mathbf{z}_t' - \mathbf{z}_t^*||_2^2 + 2||\boldsymbol{\theta}_{t+1} - \boldsymbol{\theta}^*||^2_{\epsilon\mathbf{I}}$$

$$= 3\left\|\frac{\mathbf{z}_t + 2\mathbf{z}_t'}{3} - \mathbf{z}_t^*\right\|_2^2 - \frac{1}{3}||\mathbf{z}_t + 2\mathbf{z}_t'||_2^2 + ||\mathbf{z}_t||_2^2 + 2||\mathbf{z}_t'||_2^2 + 2||\boldsymbol{\theta}_{t+1} - \boldsymbol{\theta}^*||^2_{\epsilon\mathbf{I}}$$

$$= 3\left\|\frac{\mathbf{z}_t + 2\mathbf{z}_t'}{3} - \mathbf{z}_t^*\right\|_2^2 + \frac{2}{3}||\mathbf{z}_t - \mathbf{z}_t'||_2^2 + 2||\boldsymbol{\theta}_{t+1} - \boldsymbol{\theta}^*||^2_{\epsilon\mathbf{I}} \ .$$

Let $\bar{\mathbf{z}}_s = (\mathbf{z}_s + 2\mathbf{z}_s')/3$. Then,

$$||\bar{\mathbf{z}}_t - \mathbf{z}_t^*||_2^2 + ||\boldsymbol{\theta}_{t+1} - \boldsymbol{\theta}^*||^2_{(2\epsilon/3)\mathbf{I}} \le -\frac{2}{9}||\mathbf{z}_t - \mathbf{z}_t'||_2^2 + \frac{1}{3} + \frac{4}{3\kappa}B_t^{\text{ONS}}$$

$$+ \frac{8R^2}{3\kappa^2}\log\left(\frac{2}{\delta}\sqrt{1 + \frac{2}{\kappa}B_t^{\text{ONS}} + \frac{4R^4}{\kappa^4\delta^2}}\right)$$

We lower bound the LHS with $||\bar{\mathbf{z}}_t - \mathbf{z}_t^*||_2^2 + \frac{2\epsilon}{3}||\boldsymbol{\theta}^*||_2^2 - \frac{4\epsilon}{3}S^2 =: D_2$. Define $\overline{\mathbf{W}}_t := \mathbf{X}_t^\top\mathbf{X}_t + (2\epsilon/3)\mathbf{I}$ and $\widehat{\boldsymbol{\theta}}_t^+ = \overline{\mathbf{W}}_t^{-1}\mathbf{X}_t^\top\bar{\mathbf{z}}_t$. Then,

$$D_2 = ||\boldsymbol{\theta}^* - \widehat{\boldsymbol{\theta}}_t^+||^2_{\overline{\mathbf{W}}_t} + \underbrace{||\bar{\mathbf{z}}_t - \mathbf{X}_t\widehat{\boldsymbol{\theta}}_t^+||_2^2 + \frac{2\epsilon}{3}||\widehat{\boldsymbol{\theta}}_t^+||_2^2}_{=||\bar{\mathbf{z}}_t||_2^2 - \bar{\mathbf{z}}_t^\top\mathbf{X}_t\widehat{\boldsymbol{\theta}}_t^+} - \frac{4\epsilon}{3}S^2 \ .$$

Thus,

$$||\boldsymbol{\theta}^* - \widehat{\boldsymbol{\theta}}_t^+||^2_{\overline{\mathbf{W}}_t} \le -||\bar{\mathbf{z}}_t||_2^2 + \bar{\mathbf{z}}_t^\top\mathbf{X}_t\widehat{\boldsymbol{\theta}}_t^+ + \frac{4\epsilon}{3}S^2 - \frac{2}{9}||\mathbf{z}_t - \mathbf{z}_t'||_2^2 + \frac{1}{3} + \frac{4}{3\kappa}B_t^{\text{ONS}}$$

$$+ \frac{8R^2}{3\kappa^2}\log\left(\frac{2}{\delta}\sqrt{1 + \frac{2}{\kappa}B_t^{\text{ONS}} + \frac{4R^4}{\kappa^4\delta^2}}\right)$$

$$=: \beta_t^{\text{ONS}+} \ .$$

This leads to the following confidence set:

$$C_t^{\text{ONS}+} = \{\boldsymbol{\theta} \in \mathbb{R}^d : ||\boldsymbol{\theta} - \widehat{\boldsymbol{\theta}}_t^+||^2_{\overline{\mathbf{W}}_t} \le \beta_t^{\text{ONS}+}\} \ .$$

## F Details on GLOC-TS

For simplicity, we present the algorithm and the analysis when GLOC-TS is combined with the confidence set $C_t^{\text{ONS}}$. To present the algorithm, we use the definition $\beta_t^{\text{ONS}}$ from Corollary 1. We use the notation $\beta_t^{\text{ONS}}(\delta)$ to show the dependence on $\delta$ explicitly. We present the algorithm of GLOC-TS in Algorithm 3.

---

**Algorithm 3** GLOC-TS (GLOC - Thompson Sampling)

---

1: **Input**: time horizon $T$, $\delta \in (0,1)$, $\lambda > 0$, $S > 0$, $\kappa > 0$.
2: Let $\delta' = \delta/(8T)$.
3: **for** $t = 1, 2, \ldots, T$ **do**
4:     Sample $\boldsymbol{\xi}_t \sim \mathcal{N}(0, \mathbf{I})$.
5:     Compute parameter $\dot{\boldsymbol{\theta}}_t = \widehat{\boldsymbol{\theta}}_{t-1} + \sqrt{\beta_{t-1}^{\text{ONS}}(\delta')} \cdot \overline{\mathbf{V}}_{t-1}^{-1/2} \boldsymbol{\xi}_t$
6:     Solve $\mathbf{x}_t = \arg\max_{\mathbf{x} \in \mathcal{X}} \mathbf{x}^\top \dot{\boldsymbol{\theta}}_t$.
7:     Pull $\mathbf{x}_t$ and then observe $y_t$.
8:     Feed into $\mathcal{B}$ the loss function $\ell_t(\boldsymbol{\theta}) = \ell(\mathbf{x}_t^\top \boldsymbol{\theta}, y_t)$.
9:     Update $\widehat{\boldsymbol{\theta}}_t$ and $\overline{\mathbf{V}}_t$.
10: **end for**

---

Define $\gamma_t(\delta) = \beta_t(\delta) 2d \log(2d/\delta)$ and $p = \frac{1}{4\sqrt{e\pi}}$. We present the full statement of Theorem 4 as follows.

**Theorem 4** Let $\delta' = \delta/(8T)$. The cumulative regret of GLOC-TS over $T$ steps is bounded as, w.p. at least $1 - \delta$,

$$\text{Regret}_T \leq L\left(\sqrt{\beta_T(\delta')} + \sqrt{\gamma_T(\delta')}(1 + 2/p)\right)\sqrt{2Td\log(1 + T/\lambda)} + \frac{2L}{p}\sqrt{\gamma_T(\delta')}\sqrt{\frac{8T}{\lambda}\log(4/\delta)}$$

$$= O\left(\frac{L(L+R)}{\kappa}d^{3/2}\sqrt{\log(d)T}\log^{3/2}T\right)$$

*Proof.* Note that the proof is a matter of realizing that the proof of Abeille & Lazaric [3, Lemma 4] relies on a given confidence set $C_t$ in a form of (6). To be specific, notice that the notations $k_\mu$, $c_\mu$, and $\beta_t(\delta')$ used in Abeille & Lazaric [3] is equivalent to $L$, $\kappa$, and $\kappa\sqrt{\beta_t(\delta)}$ in this paper, respectively. Furthermore, our result stated in (8) can replace the result of Abeille & Lazaric [3, Prop. 11]. Finally, one can verify that the proof of Abeille & Lazaric [3, Lemma 4] leads to the proof of the theorem above. $\square$

Although the theorem above is stated under the fixed-buget setting, one can easily change the failure rate $\delta'$ to $O(t^2/\delta)$ and enjoy an anytime regret bound.

## G QGLOC without $r$

We present an alternative definition of $m_t$ that does not rely on $r$:

$$\overline{m}'_{t-1} = ||\mathbf{x}^{\text{Greedy}}||_{\overline{\mathbf{V}}_{t-1}^{-1}}, \text{ where } \mathbf{x}^{\text{Greedy}} := \arg\max_{\mathbf{x} \in \mathcal{X}_{t-1}} \langle \widehat{\boldsymbol{\theta}}_{t-1}, \mathbf{x} \rangle$$

We claim that $\overline{m}'_{t-1} \leq ||\mathbf{x}_t^{\text{QGLOC}}||_{\overline{\mathbf{V}}_{t-1}^{-1}}$. To see this, suppose not: $\overline{m}'_{t-1} > ||\mathbf{x}_t^{\text{QGLOC}}||_{\overline{\mathbf{V}}_{t-1}^{-1}}$. Then, $\mathbf{x}^{\text{Greedy}} \neq \mathbf{x}^{\text{QGLOC}}$. Furthermore, $\mathbf{x}^{\text{Greedy}}$ must be the maximizer of the QGLOC objective function while $\mathbf{x}^{\text{Greedy}} \neq \mathbf{x}^{\text{QGLOC}}$, which is a contradiction. The claim above allows all the proofs of our theorems on QGLOC to go through with $\overline{m}'_{t-1}$ in place of $\overline{m}_{t-1}$.

However, computing $\overline{m}'_{t-1}$ requires another hashing for finding the greedy arm defined above. Although this introduces a factor of 2 in the time complexity, it is quite cumbersome in practice, which is why we stick to $m_{t-1}$ defined in (9) in the main text.

# H   Proof of Theorem 5

Let us first present the background. Denote by $x_t^{\text{GLOC}}$ the solution of the optimization problem at line 3 of Algorithm 1. To find $x_t^{\text{GLOC}}$, one can fix $\mathbf{x}$ and find the maximizer $\tilde{\boldsymbol{\theta}}(\mathbf{x}) := \max_{\boldsymbol{\theta} \in C_{t-1}} \mathbf{x}^\top \boldsymbol{\theta}$ in a closed form using the Lagrangian method:

$$\tilde{\boldsymbol{\theta}}(\mathbf{x}) = \widehat{\boldsymbol{\theta}}_{t-1} + \sqrt{\beta_{t-1}} \cdot \frac{\overline{\mathbf{V}}_{t-1}^{-1} \mathbf{x}}{||\mathbf{x}||_{\overline{\mathbf{V}}_{t-1}^{-1}}} \,, \tag{19}$$

which is how we obtained (7).

The following lemma shows that the objective function (10) plus $c_0 \beta^{3/4} \overline{m}_{t-1}$ is an upper bound of the GLOC's objective function (7). Note that this holds for any sequence $\{\beta_t\}$.

**Lemma 5.**

$$\langle \widehat{\boldsymbol{\theta}}_{t-1}, \mathbf{x} \rangle + \sqrt{\beta_{t-1}} ||\mathbf{x}||_{\overline{\mathbf{V}}_{t-1}^{-1}}$$

$$\leq \langle \widehat{\boldsymbol{\theta}}_{t-1}, \mathbf{x} \rangle + \frac{\beta_{t-1}^{1/4}}{4 c_0 \overline{m}_{t-1}} \cdot ||\mathbf{x}||_{\overline{\mathbf{V}}_{t-1}^{-1}}^2 + c_0 \beta_{t-1}^{3/4} \overline{m}_{t-1} \,.$$

*Furthermore,*

$$\langle \widehat{\boldsymbol{\theta}}_{t-1}, \mathbf{x}_t^{\text{GLOC}} \rangle + \sqrt{\beta_{t-1}} ||\mathbf{x}_t^{\text{GLOC}}||_{\overline{\mathbf{V}}_{t-1}^{-1}}$$

$$\leq \langle \widehat{\boldsymbol{\theta}}_{t-1}, \mathbf{x}_t^{\text{QGLOC}} \rangle + \frac{\beta_{t-1}^{1/4}}{4 c_0 \overline{m}_{t-1}} ||\mathbf{x}_t^{\text{QGLOC}}||_{\overline{\mathbf{V}}_{t-1}^{-1}}^2 + c_0 \beta_{t-1}^{3/4} \overline{m}_{t-1} \,. \tag{20}$$

*Proof.* Recall the GLOC optimization problem defined at line 3 of Algorithm 1. For a fixed $\mathbf{x}$, we need to solve

$$\tilde{\boldsymbol{\theta}}(\mathbf{x}) = \arg\min_{\boldsymbol{\theta}} \quad - \langle \boldsymbol{\theta}, \mathbf{x} \rangle$$

$$\text{s.t.} \quad ||\boldsymbol{\theta} - \widehat{\boldsymbol{\theta}}_{t-1}||_{\overline{\mathbf{V}}_{t-1}}^2 - \beta_{t-1} \leq 0$$

The Lagrangian is $\mathcal{L}(\boldsymbol{\theta}, \tau) := -\langle \boldsymbol{\theta}, \mathbf{x} \rangle + \tau(||\boldsymbol{\theta} - \widehat{\boldsymbol{\theta}}_{t-1}||_{\overline{\mathbf{V}}_{t-1}}^2 - \beta_{t-1})$, and thus we need to solve

$$\max_{\tau \geq 0} \min_{\boldsymbol{\theta}} \quad \mathcal{L}(\boldsymbol{\theta}, \tau) \,.$$

According to the first-order optimality condition,

$$-\frac{\partial \mathcal{L}(\boldsymbol{\theta}, \tau)}{\partial \boldsymbol{\theta}} = \mathbf{x} - \tau(2\overline{\mathbf{V}}_{t-1}\boldsymbol{\theta} - 2\overline{\mathbf{V}}_{t-1}\widehat{\boldsymbol{\theta}}_{t-1}) = 0$$

$$\boldsymbol{\theta} = \widehat{\boldsymbol{\theta}}_{t-1} + (2\tau)^{-1}\overline{\mathbf{V}}_{t-1}^{-1}\mathbf{x} \,, \tag{21}$$

which results in $(\min_{\boldsymbol{\theta}} \mathcal{L}(\boldsymbol{\theta}, \tau)) = -\langle \widehat{\boldsymbol{\theta}}_{t-1}, \mathbf{x} \rangle - (4\tau)^{-1}||\mathbf{x}||_{\overline{\mathbf{V}}_{t-1}^{-1}}^2 - \tau \beta_{t-1}$. It remains to solve

$$\max_{\tau \geq 0} \quad -\langle \hat{\boldsymbol{\theta}}, \mathbf{x} \rangle - (4\tau)^{-1}||\mathbf{x}||_{\overline{\mathbf{V}}_{t-1}^{-1}}^2 - \tau \beta_{t-1},$$

whose first-order optimality says that the solution is $\tau_* := (2\sqrt{\beta_{t-1}})^{-1}||\mathbf{x}||_{\overline{\mathbf{V}}^{-1}}$. Plugging $\tau \leftarrow \tau_*$ in (21) leads to the solution $\tilde{\boldsymbol{\theta}}(\mathbf{x})$ defined in (19). Note that any choice of $\tau \geq 0$ leads to a lower bound on the Lagrangian $\mathcal{L}(\tilde{\boldsymbol{\theta}}(\mathbf{x}), \tau)$ . Define

$$\tilde{\tau} := c_0 \beta_{t-1}^{-1/4} \overline{m}_{t-1} \,.$$

Then,

$$\mathcal{L}(\tilde{\boldsymbol{\theta}}(\mathbf{x}), \tilde{\tau}) = -\langle \widehat{\boldsymbol{\theta}}_{t-1}, \mathbf{x} \rangle - \frac{\beta_{t-1}^{1/4}}{4 c_0 \overline{m}_{t-1}} ||\mathbf{x}||_{\overline{\mathbf{V}}_{t-1}^{-1}}^2 - c_0 \beta_{t-1}^{3/4} \overline{m}_{t-1}$$

$$\leq \mathcal{L}(\tilde{\boldsymbol{\theta}}(\mathbf{x}), \tau_*) = -\langle \widehat{\boldsymbol{\theta}}_{t-1}, \mathbf{x} \rangle - \sqrt{\beta_{t-1}} ||\mathbf{x}||_{\overline{\mathbf{V}}_{t-1}^{-1}} \,.$$

This concludes the first part of the lemma. The second part of the lemma trivially follows from the first part by the definition (10). $\square$

*Proof.* Assume $\boldsymbol{\theta}^* \in C_t^{\text{ONS}}$ for all $t \geq 1$, which happens w.p. at least $1 - \delta$. Suppose we pull $\mathbf{x}_t^{\text{QGLOC}}$ at every iteration. While we use the confidence set $C_t^{\text{ONS}}$, we omit the superscript ONS from $\beta_t^{\text{ONS}}$ for brevity. Recall that $\bar{\beta}_t$ is an upper bound on $\beta_t$ that is nondecreasing in $t$. We bound the instantaneous regret $r_t$ as follows:

$$
\begin{aligned}
\frac{r_t}{L} &= \frac{1}{L}\left(\mu(\langle \boldsymbol{\theta}^*, \mathbf{x}_{t,*}\rangle) - \mu(\langle \boldsymbol{\theta}^*, \mathbf{x}_t^{\text{QGLOC}}\rangle)\right) \\
&\leq (\langle \boldsymbol{\theta}^*, \mathbf{x}_{t,*}\rangle - \langle \boldsymbol{\theta}^*, \mathbf{x}_t^{\text{QGLOC}}\rangle) \\
&\leq \langle \tilde{\boldsymbol{\theta}}(\mathbf{x}_t^{\text{GLOC}}), \mathbf{x}_t^{\text{GLOC}}\rangle - \langle \boldsymbol{\theta}^*, \mathbf{x}_t^{\text{QGLOC}}\rangle \\
&= \langle \widehat{\boldsymbol{\theta}}_{t-1}, \mathbf{x}_t^{\text{GLOC}}\rangle + \sqrt{\bar{\beta}_{t-1}}\|\mathbf{x}_t^{\text{GLOC}}\|_{\overline{\mathbf{V}}_{t-1}^{-1}} - \langle \boldsymbol{\theta}^*, \mathbf{x}_t^{\text{QGLOC}}\rangle \\
&\overset{\text{(Lem. 5)}}{\leq} \langle \widehat{\boldsymbol{\theta}}_{t-1}, \mathbf{x}_t^{\text{QGLOC}}\rangle + \frac{\bar{\beta}_{t-1}^{1/4}}{4c_0 \overline{m}_{t-1}}\|\mathbf{x}_t^{\text{QGLOC}}\|_{\overline{\mathbf{V}}_{t-1}^{-1}}^2 + c_0 \bar{\beta}_{t-1}^{3/4}\overline{m}_{t-1} - \langle \boldsymbol{\theta}^*, \mathbf{x}_t^{\text{QGLOC}}\rangle \\
&= \langle \widehat{\boldsymbol{\theta}}_{t-1} - \boldsymbol{\theta}^*, \mathbf{x}_t^{\text{QGLOC}}\rangle + \frac{\bar{\beta}_{t-1}^{1/4}}{4c_0 \overline{m}_{t-1}}\|\mathbf{x}_t^{\text{QGLOC}}\|_{\overline{\mathbf{V}}_{t-1}^{-1}}^2 + c_0 \bar{\beta}_{t-1}^{3/4}\overline{m}_{t-1} \\
&\leq \underbrace{\|\widehat{\boldsymbol{\theta}}_{t-1} - \boldsymbol{\theta}^*\|_{\overline{\mathbf{V}}_{t-1}}\|\mathbf{x}_t^{\text{QGLOC}}\|_{\overline{\mathbf{V}}_{t-1}^{-1}}}_{=:A_1(t)} + \underbrace{\frac{\bar{\beta}_{t-1}^{1/4}}{4c_0 \overline{m}_{t-1}}\|\mathbf{x}_t^{\text{QGLOC}}\|_{\overline{\mathbf{V}}_{t-1}^{-1}}^2}_{:=A_2(t)} + \underbrace{c_0 \bar{\beta}_{t-1}^{3/4}\overline{m}_{t-1}}_{=:A_3(t)} \;.
\end{aligned}
$$

Note that $\langle \boldsymbol{\theta}^*, \mathbf{x}\rangle \in [-S, S]$ implies that $r_t \leq 2LS$. Then,

$$
\begin{aligned}
r_t &\leq \min\{2LS, L \cdot A_1(t) + L \cdot A_2(t) + L \cdot A_3(t)\} \\
&\leq L\min\{2S, A_1(t)\} + \min\{2S, A_2(t)\} + \min\{2S, A_3(t)\} \;.
\end{aligned}
$$

where the last inequality can be shown by a case-by-case analysis on each $\min$ operator.

Now, we consider computing $\sum_{t=1}^{T} r_t$. Using the same argument as the proof of Theorem 2 and Corollary 2,

$$
L\sum_{t=1}^{T}\min\{2S, A_1(t)\} = O\left(\frac{L(L+R)}{\kappa}d\sqrt{T}\log^{3/2}(T)\right) \;.
$$

Then,

$$
\begin{aligned}
&L\sum_{t=1}^{T}\min\left\{2S, \frac{\bar{\beta}_{T-1}^{1/4}}{4c_0\overline{m}_{t-1}}\|\mathbf{x}_t^{\text{QGLOC}}\|_{\overline{\mathbf{V}}_{t-1}^{-1}}^2\right\} \\
&\leq L\sum_{t=1}^{T}\min\left\{2S, \frac{\bar{\beta}_{T-1}^{1/4}}{4c_0 r}\sqrt{T+\lambda}\|\mathbf{x}_t^{\text{QGLOC}}\|_{\overline{\mathbf{V}}_{t-1}^{-1}}^2\right\} \\
&\leq \frac{L\bar{\beta}_{T-1}^{1/4}}{4c_0 r}\sqrt{T+\lambda}\sum_{t=1}^{T}\min\left\{\frac{8c_0 rS}{\bar{\beta}_{T-1}^{1/4}\sqrt{T+\lambda}}, \|\mathbf{x}_t^{\text{QGLOC}}\|_{\overline{\mathbf{V}}_{t-1}^{-1}}^2\right\} \\
&\overset{\text{(Lem. 3)}}{\leq} \frac{L\bar{\beta}_{T-1}^{1/4}}{4c_0 r}\sqrt{T+\lambda}\max\left\{2, \frac{8c_0 rS}{\bar{\beta}_{T-1}^{1/4}\sqrt{T+\lambda}}\right\}\sum_{t=1}^{T}\log(1+\|\mathbf{x}_t^{\text{QGLOC}}\|_{\overline{\mathbf{V}}_{t-1}^{-1}}^2) \\
&\overset{(15)}{=} L\frac{1}{c_0}\cdot O\left(\left(\frac{L^2+R^2}{\kappa^2}d\log^2 T\right)^{1/4}\right)\cdot O(\sqrt{T})\cdot O(d\log T) \\
&= O\left(\frac{1}{c_0}L\left(\frac{L+R}{\kappa}\right)^{1/2}d^{5/4}\sqrt{T}\log^{3/2}T\right)
\end{aligned}
$$

and

$$
L\sum_{t=1}^{T}\min\{2S, c_0\bar{\beta}_{t-1}^{3/4}\overline{m}_{t-1}\}
$$

$$\overset{(9)}{\leq} L \sum_{t=1}^{T} \min\{2S, c_0 \bar{\beta}_{T-1}^{3/4} ||\mathbf{x}_t^{\text{QGLOC}}||_{\overline{\mathbf{V}}_{t-1}^{-1}}\}$$

$$\leq L c_0 \bar{\beta}_{T-1}^{3/4} \sum_{t=1}^{T} \min\left\{\frac{2S}{c_0 \bar{\beta}_{T-1}^{3/4}}, ||\mathbf{x}_t^{\text{QGLOC}}||_{\overline{\mathbf{V}}_{t-1}^{-1}}\right\}$$

$$\overset{(a)}{\leq} L c_0 \bar{\beta}_{T-1}^{3/4} \sqrt{T \sum_{t=1}^{T} \min\left\{\left(\frac{2S}{c_0 \bar{\beta}_{T-1}^{3/4}}\right)^2, ||\mathbf{x}_t^{\text{QGLOC}}||_{\overline{\mathbf{V}}_{t-1}^{-1}}^2\right\}}$$

$$\overset{(\text{Lem. 3})}{\leq} L c_0 \bar{\beta}_{T-1}^{3/4} \sqrt{T \max\left\{2, \left(\frac{2S}{c_0 \bar{\beta}_{T-1}^{3/4}}\right)^2\right\} \sum_{t=1}^{T} \log(1 + ||\mathbf{x}_t^{\text{QGLOC}}||_{\overline{\mathbf{V}}_{t-1}^{-1}}^2)}$$

$$\overset{(15)}{=} c_0 L \cdot O\left(\left(\frac{L^2 + R^2}{\kappa^2} d \log^2 T\right)^{3/4}\right) \cdot O(\sqrt{Td \log T})$$

$$= O\left(c_0 L \left(\frac{L+R}{\kappa}\right)^{3/2} d^{5/4} \sqrt{T} \log^2 T\right) ,$$

where $(a)$ is due to the Cauchy-Schwartz inequality. Therefore, the regret is $O\left(L(\frac{1}{c_0} \left(\frac{L+R}{\kappa}\right)^{1/2} + c_0 \left(\frac{L+R}{\kappa}\right)^{3/2}) d^{5/4} \sqrt{T} \log^2(T)\right)$. One can see that setting $c_0 = c_0' \left(\frac{L+R}{\kappa}\right)^{-1/2}$ leads to the stated regret bound. Note that one can improve $\log^2(T)$ in the regret bound to $\log^{7/4}(T)$ by making $c_0$ scale with $\log^{-1/4}(t)$, which is left as an exercise.

When the noise $|\eta_t|$ is bounded, we have a tighter $\beta_t$ and thus we can replace $\log^2(T)$ in the regret bound to $\log^{5/4}(T)$. $\qquad\qquad\square$

# I  On $c_0$ of QGLOC

Observe that in (10) $c_0$ is a free parameter that adjusts the balance between the exploitation (the first term) and exploration (the second term). This is an interesting characteristic that is not available in existing algorithms but is attractive to practitioners. Specifically, in practice existing bandit algorithms like OFUL [1], LTS [5], and others [15, 41] usually perform exploration more than necessary, so one often enforces more exploitation by multiplying a small constant less than 1 to $\sqrt{\beta_t}$; e.g., see [40, 8]. Applying such a trick is theoretically not justified and foregoes the regret guarantee for existing algorithms, so a practitioner must take a leap of faith. In contrast, adjusting $c_0$ of QGLOC is exactly the common heuristic but now does not break the regret guarantee, which can assure practitioners.

# J  Details on Hashing

We first briefly introduce hashing methods for fast similarity search. Here, the similarity measure is often Euclidean distance [12] or inner product [35]. For a comprehensive review, we refer to Wang et al. [39]. The hashing methods build a hash table that consists of buckets where each bucket is identified by a unique hash key that is a sequence of $k$ integers. At the hashing construction time, for each data point $\mathbf{x}$ in the database we compute its hash key $h(\mathbf{x})$ using a function $h$ (details shown below). We then organize the hash table so that each bucket contains pointers to the actual data points with the same hash key. The hash functions are decided at the construction time. Typically, for $d'$-dimensional hashing one draws $k$ independent normally-distributed $d'$-dimensional vectors, which we call *projection vectors*. The hash function $h(\mathbf{x})$ outputs a discretized version of the inner product between these $k$ vectors and the data point $\mathbf{x}$. When processing a query $\mathbf{q}$, we compute the hash key $h(\mathbf{q})$, retrieve the corresponding bucket, compute the similarities between the query and the data points therein, and pick the most similar one. It is important that one uses the same projection vectors for constructing the table and determining the hash key of the query. This means one needs to store the projection vectors. Finally, one typically constructs $U$ independent hash tables to reduce the chance of missing very similar points (i.e., to increase the recall).

We now turn to operating QGLOC with hashing. Note that one would like to use an accurate hashing scheme since how accurately one solves (10) can impact the regret. However, a more accurate hashing have a higher space and time complexity. We characterize such a tradeoff in this section.

Recall that we aim to guarantee the accuracy of a MIPS hashing by the $c_{\mathrm{H}}$-MIPS-ness. As we have mentioned in the main text, however, existing MIPS algorithms do not directly offer a $c_{\mathrm{H}}$-MIPS guarantee. Instead of the $c_{\mathrm{H}}$-MIPS guarantee, the standard MIPS algorithms provide a less convenient guarantee for an input parameter $c$ and $M$ as follows:

**Definition 2.** *Let $\mathcal{X} \subseteq \mathbb{R}^{d'}$ such that $|\mathcal{X}| < \infty$. A data point $\tilde{\mathbf{x}} \in \mathcal{X}$ is called $(c, M)$-MIPS w.r.t. a given query $\mathbf{q}$ if it satisfies $\langle \mathbf{q}, \tilde{\mathbf{x}} \rangle \geq cM$. An algorithm is called $(c, M)$-MIPS if, given a query $\mathbf{q} \in \mathbb{R}^{d'}$, it retrieves $\mathbf{x} \in \mathcal{X}$ that is $(c, M)$-MIPS w.r.t. $\mathbf{q}$ whenever there exists $\mathbf{x}' \in \mathcal{X}$ such that $\langle \mathbf{q}, \mathbf{x}' \rangle \geq M$.*

Note that when there is no such $\mathbf{x}$ that $\langle \mathbf{q}, \mathbf{x} \rangle \geq M$, retrieving any arbitrary vector in $\mathcal{X}$ is qualified as being $(c, M)$-MIPS. This also means that, by its contrapositive, if the hashing returns a vector that is not $(c, M)$-MIPS w.r.t. $\mathbf{q}$, then there is no $\mathbf{x}$ such that $\langle \mathbf{q}, \mathbf{x} \rangle \geq M$ with high probability.

We emphasize that, to enjoy the $(c, M)$-MIPS-ness, a hashing must be built based on $c$ and $M$. If we know $\max_{\mathbf{x} \in \mathcal{X}_t} \langle \mathbf{q}_t, \mathbf{x} \rangle$ ahead of time, then we can just set $M = \max_{\mathbf{x} \in \mathcal{X}_t} \langle \mathbf{q}_t, \mathbf{x} \rangle$ and build a hashing that is $(c, M)$-MIPS for some $c$, which gives a $c$-MIPS guarantee for the query $\mathbf{q}_t$. However, one does not have such information, and each query $\mathbf{q}_t$ has its own "useful" value $M$.

To overcome such a difficulty, it seems natural to construct a $c$-MIPS hashing using multiple $(c, M)$-MIPS hashings with various $M$ values covering a wide range. Indeed, such a construction is described in [18] for locality-sensitive hashing based on the Euclidean distance, which is complicated by multiple subroutines. We present here a streamlined construction of a $c_{\mathrm{H}}$-MIPS hashing thanks to the existence of a high-probability upper and lower bound on the maximum of the QGLOC objective (10) as we show below. Note that one can derive a similar result for GLOC-TS.

Hereafter, we omit ONS from $\beta_t^{\mathrm{ONS}}$. Note that there exists a simple upper bound $\overline{\beta}_t$ on $\beta_t$ (see the proof of Corollary 2). Define $E_1(\delta)$ to be the event that $\boldsymbol{\theta}^*$ belongs to $C_t$ for all $t \geq 1$.

**Lemma 6.** *Assume $E_1(\delta)$ and $\max_{\mathbf{x} \in \mathcal{X}_t} \langle \boldsymbol{\theta}^*, \mathbf{x} \rangle \geq 1/2$. Suppose the target time horizon $T$ is given. Then,*

$$M_{\min} := 1/2 \leq \max_{t \in [T]} \max_{\mathbf{x} \in \mathcal{X}_t} \langle \widehat{\boldsymbol{\theta}}_{t-1}, \mathbf{x} \rangle + \frac{\beta_{t-1}^{1/4}}{4c_0 \overline{m}_{t-1}} ||\mathbf{x}||_{\mathbf{V}_{t-1}^{-1}}^2$$

$$\leq \sqrt{d}S + \overline{\beta}_{T-1}^{1/4} \cdot \frac{\sqrt{T + \lambda}}{4c_0 r \lambda} =: M_{\max} .$$

Before presenting the proof, note that Lemma 6 assumes that $\max_{\mathbf{x} \in \mathcal{X}_t} \langle \boldsymbol{\theta}^*, \mathbf{x} \rangle \geq 1/2$. In practice, this is not a restrictive assumption since it means that there exists at least one arm for which the reward is decently large in expectation. In interactive retrieval systems with binary user feedback, for example, it is reasonable to assume that there exists at least one item to which the user is likely to give a positive reward since otherwise any algorithm would work almost equally badly. One can change $1/2$ to any reasonable number $v$ for which the reward $\mu(v)$ is considered high.

*Proof.* To show the lowerbound, recall that the objective function (10) is derived as an upperbound of the original GLOC objective function (7). Since the original GLOC objective function has $\boldsymbol{\theta}^*$ as a feasible point by $E_1(\delta)$, $\max_{\mathbf{x} \in \mathcal{X}_t} \langle \boldsymbol{\theta}^*, \mathbf{x} \rangle$ becomes a trivial lowerbound of the maximum of the GLOC objective and also of the maximum of QGLOC objective (10). This proves the lowerbound of the Lemma.

For the remaining part of the proof, we use notation $\mathbf{X}, \boldsymbol{\eta}, \mathbf{z}$ and $\overline{\mathbf{V}}$ in place of $\mathbf{X}_t, \boldsymbol{\eta}_t, \mathbf{z}_t$, and $\overline{\mathbf{V}}_t$ respectively (recall that $\mathbf{z}$ is defined in Section 3).

It is easy to see that the eigenvalues of $\mathbf{X}\overline{\mathbf{V}}^{-2}\mathbf{X}^\top$ are all less than 1 (use the SVD of $\mathbf{X}$). Furthermore, $z_s = \mathbf{x}_s^\top \boldsymbol{\theta}_s \in [-S, S]$, and so $||\mathbf{z}||_2 \leq \sqrt{d}S$. Using these facts,

$$\begin{aligned}
||\widehat{\boldsymbol{\theta}}_{t-1}||_2 &= ||\overline{\mathbf{V}}^{-1}\mathbf{X}^\top \mathbf{z}||_2 \\
&= \sqrt{(\mathbf{z})^\top \mathbf{X}\overline{\mathbf{V}}^{-2}\mathbf{X}^\top \mathbf{z}} \\
&= ||(\mathbf{X}\overline{\mathbf{V}}^{-2}\mathbf{X}^\top)^{1/2}\mathbf{z}||_2 \\
&\leq ||\mathbf{z}||_2 \leq \sqrt{d}S .
\end{aligned}$$

Finally,

$$\begin{aligned}
\max_{t \in [T]} \max_{\mathbf{x} \in \mathcal{X}_t} \langle \widehat{\boldsymbol{\theta}}_{t-1}, \mathbf{x}\rangle + \frac{\beta_{t-1}^{1/4}}{4c_0 \overline{m}_{t-1}}||\mathbf{x}||^2_{\overline{\mathbf{V}}_{t-1}^{-1}} &\leq \max_{t \in [T]} \max_{\mathbf{x} \in \mathcal{X}_t} ||\widehat{\boldsymbol{\theta}}_{t-1}||_2 + \frac{\beta_{t-1}^{1/4}}{4c_0 \overline{m}_{t-1}}||\mathbf{x}||^2_{\overline{\mathbf{V}}_{t-1}^{-1}} \\
&\leq \max_{t \in [T]} \sqrt{d}S + \beta_{t-1}^{1/4} \cdot \frac{\sqrt{t+\lambda}}{4c_0 r} \cdot \frac{1}{\lambda} \\
&\leq \sqrt{d}S + \overline{\beta}_{T-1}^{1/4} \cdot \frac{\sqrt{T+\lambda}}{4c_0 r} \cdot \frac{1}{\lambda} .
\end{aligned}$$

$\square$

Given a target approximation level $c_{\mathrm{H}} < 1$, we construct a $c_{\mathrm{H}}$-MIPS as follows. Define

$$J := \left\lceil \log_{1/\sqrt{c_{\mathrm{H}}}} \frac{M_{\max}}{M_{\min}} \right\rceil = O(\log(dT)/\log(c_{\mathrm{H}}^{-1})) . \tag{22}$$

We build a series of $J$ MIPS hashing schemes that are

$$(c_{\mathrm{H}}^{1/2}, c_{\mathrm{H}}^{j/2} M_{\max})\text{-MIPS for } j \in [J] . \tag{23}$$

We say that the MIPS hashing *succeeds* (*fails*) for a query $\mathbf{q}$ if the retrieved vector is (not) $(c, M)$-MIPS w.r.t. $\mathbf{q}$. Theorem 6 shows that one can perform a binary search to find a vector $\mathbf{x} \in \mathcal{X}$ that is $c_{\mathrm{H}}$-MIPS.

**Theorem 6.** *Upon given a query $\mathbf{q}$, perform a binary search over the $J$ MIPS hashings* (23) *to find the smallest $j^* \in [J]$ for which the retrieved vector $\mathbf{x}^{(j^*)}$ from the $j^*$-th hashing succeeds. Then, $\mathbf{x}^{(j^*)}$ is $c_{\mathrm{H}}$-MIPS w.r.t. $\mathbf{q}$ with high probability.*

*Proof.* Assume the event that a retrieved vector $\mathbf{x}^{(j)}$ from $j$-th MIPS satisfies $(c_{\mathrm{H}}^{1/2}, c_{\mathrm{H}}^{j/2} M_{\max})$-MIPS for $j \in [J]$, which happens with high probability.

Define the maximum $M^* := \max_{\mathbf{x} \in \mathcal{X}_t} \langle \mathbf{q}, \mathbf{x}\rangle$. The result of the binary search is that, for query $\mathbf{q}$, $j^*$-th MIPS succeeds but $(j^* - 1)$-th MIPS fails. By the definition of $(c_{\mathrm{H}}^{1/2}, c_{\mathrm{H}}^{(j^*-1)/2} M_{\max})$-MIPS, the fact that $(j^* - 1)$-th hashing fails implies $M^* < c_{\mathrm{H}}^{(j^*-1)/2} M_{\max}$. Then,

$$\langle \mathbf{q}, \mathbf{x}^{(j^*)}\rangle \geq c_{\mathrm{H}}^{1/2} \cdot c_{\mathrm{H}}^{j^*} M_{\max} = c_{\mathrm{H}} \cdot c_{\mathrm{H}}^{(j^*-1)/2} M_{\max} > c_{\mathrm{H}} M^* .$$

$\square$

Among various $(c, M)$-MIPS algorithms [35, 36, 30, 17], we adopt Shrivastava *et al.* [35]. Shrivastava et al. propose a reduction of MIPS to locality-sensitive hashing and present a result that their algorithm is $(c, M)$-MIPS with $O(N^{\rho^*} \log N)$ inner product computations and space $O(N^{1+\rho^*})$ for an optimized value $\rho^*$ that is guaranteed to be less than 1; see [35, Theorem 5] for detail.

Let $d'$ be the dimensionality of the projection vectors, where $d' = d^2 + d$ for QGLOC and $d' = d$ for GLOC-TS. One can recover the order of the space and time complexity stated in the main text by $O(\log(J)N^{\rho^*} \log(N)d')$ and $O(JN^{\rho^*}(N + \log(N)d'))$, respectively.

# K  The Regret Bound of QGLOC under Hashing Approximation Error

For most recommendation or interactive retrieval applications, it is reasonable to assume that the total number of steps $T$ is bounded by a known constant ("fixed budget" in bandit terminology) since users do not interact with the system for too long. We present the regret of QGLOC combined with

MIPS hashing in Theorem 7. The theorem states that in the fixed budget setting $T$, we can set the target approximation level $c_\mathrm{H}$ as a function of $T$ and enjoy the same order of regret as exactly solving the maximization problem (10). The proof is presented at the end of this section.

**Theorem 7.** *Let $T \geq 2$ and $c_\mathrm{H} = \left(1 + \frac{\log(T)}{\sqrt{T}}\right)^{-1}$. Suppose we run QGLOC for $T$ iterations where we invoke a $c_\mathrm{H}$-MIPS hashing algorithm $\mathcal{H}$ to approximately find the solution of (10) at each time $t \in [T]$. Denote this algorithm by $\mathrm{QGLOC}\langle\mathcal{H}\rangle$. Assume that, w.p. at least $1 - \delta$, the hashing $\mathcal{H}$ successfully retrieves a $c_\mathrm{H}$-MIPS solution for every $T$ queries made by $\mathrm{QGLOC}\langle\mathcal{H}\rangle$. Then, w.p. at least $1 - 3\delta$, $\mathrm{Regret}_T^{\mathrm{QGLOC}\langle\mathcal{H}\rangle} = \hat{O}(\kappa^{-1} L(L + R) d^{5/4} \sqrt{T} \log^{7/4}(T))$.*

Note that we have made an assumption that, w.h.p., the hashing retrieves a $c_\mathrm{H}$-MIPS solution for every $T$ queries made by the algorithm. One naive way to construct such a hashing is to build $T$ independent hashing schemes as follows, which is the first low-regret GLB algorithm with time sublinear in $N$, to our knowledge.

**Corollary 3.** *Let $T \geq 2$ and $c_\mathrm{H} = \left(1 + \frac{\log(T)}{\sqrt{T}}\right)^{-1}$. Suppose we build $T$ independent hashings where each hashing is $c_\mathrm{H}$-MIPS w.p. at least $1 - \delta/T$. Suppose we run QGLOC for $T$ iterations where at time $t$ we use $t$-th MIPS hashing to solve (10). Then, w.p. at least $1 - 3\delta$, the regret bound is $\hat{O}(\kappa^{-1} L(L + R) d^{5/4} \sqrt{T} \log^{7/4}(T))$.*

*Proof.* The probability that at least one of $T$ hashing schemes fails to output a $c_\mathrm{H}$-MIPS solution is at most $\delta$. Combining this with Theorem 7 completes the proof. □

However, it is easy to see that its space complexity is $\Omega(T)$, which is not space-efficient. Of course, a better way would be to construct one hashing scheme that is, w.h.p., $c_\mathrm{H}$-MIPS for any sequence of $T$ queries. However, this is nontrivial for the following reason. In bandit algorithms, the second query depends on the result of the first query. Here, the result of the query $\mathbf{q}_1$ is obtained based on the hash keys computed using the *projection vectors*. Now, the second query $\mathbf{q}_2$ is based on the result of querying $\mathbf{q}_1$, which means that $\mathbf{q}_2$ now correlates with the projection vectors. This breaks the *independence* of the query with the projection vectors, thus breaking the hashing guarantee. One way to get around the issue is the union bound. This is possible when there exists a finite set of possible queries, which is indeed how [18] manage to show a guarantee on such 'adaptive' queries. In our case, unfortunately, there are infinitely many possible queries, and thus the union bound does not apply easily. Resolving the issue above is of theoretical interest and left as future work.

Meanwhile, it is hardly the case that the correlation between $\mathbf{q}_2$ and the projection vectors has an malignant effect in practice. Indeed, many existing studies using hashing such as Jain et al. [22] ignore the correlation issue and do not provide any guarantee on the adaptive queries.

### K.1 Proof of Theorem 7

*Proof.* Assume that $\boldsymbol{\theta}^*$ belongs to $C_t^{\mathrm{ONS}}$, which happens w.h.p. In this proof, we drop QGLOC from $\mathbf{x}_t^{\mathrm{QGLOC}}$ and use $\mathbf{x}_t$. Denote by $\mathbf{x}_t^{\mathcal{H}}$ the solution returned by the MIPS algorithm. We omit ONS from $\beta_t^{\mathrm{ONS}}$ to avoid clutter. Then, being $c_\mathrm{H}$-MIPS guarantees that, $\forall t \in [T]$,

$$\langle \widehat{\boldsymbol{\theta}}_{t-1}, \mathbf{x}_t^{\mathcal{H}} \rangle + \frac{\beta_{t-1}^{1/4}}{4c_0 \overline{m}_{t-1}} ||\mathbf{x}_t^{\mathcal{H}}||_{\overline{\mathbf{V}}_{t-1}^{-1}}^2 \geq c_\mathrm{H} \left( \langle \widehat{\boldsymbol{\theta}}_{t-1}, \mathbf{x}_t \rangle + \frac{\beta_{t-1}^{1/4}}{4c_0 \overline{m}_{t-1}} ||\mathbf{x}_t||_{\overline{\mathbf{V}}_{t-1}^{-1}}^2 \right). \tag{24}$$

To avoid clutter, we use $\mathbf{X}$, $\mathbf{y}$, and $\boldsymbol{\eta}$ in place of $\mathbf{X}_{t-1}$, $\mathbf{y}_{t-1}$, and $\boldsymbol{\eta}_{t-1}$, respectively, when it is clear from the context. Note that, using the techniques in the proof of Theorem 2,

$$\langle \widehat{\boldsymbol{\theta}}_{t-1}, \mathbf{x}_t^{\mathcal{H}} \rangle = \langle \boldsymbol{\theta}^*, \mathbf{x}_t^{\mathcal{H}} \rangle + \langle \widehat{\boldsymbol{\theta}}_{t-1} - \boldsymbol{\theta}^*, \mathbf{x}_t^{\mathcal{H}} \rangle \leq S + \sqrt{\overline{\beta}_{t-1}} ||\mathbf{x}_t^{\mathcal{H}}||_{\overline{\mathbf{V}}_{t-1}^{-1}}. \tag{25}$$

The instantaneous regret at time $t$ divided by $L$ is

$$\frac{r_t}{L} \leq \langle \boldsymbol{\theta}^*, \mathbf{x}_{t,*} \rangle - \langle \boldsymbol{\theta}^*, \mathbf{x}_t^{\mathcal{H}} \rangle$$

$$\leq \langle \widehat{\boldsymbol{\theta}}_{t-1}, \mathbf{x}_t \rangle + \frac{\beta_{t-1}^{1/4}}{4c_0 \overline{m}_{t-1}} ||\mathbf{x}_t||_{\overline{\mathbf{V}}_{t-1}^{-1}}^2 + c_0 \beta_{t-1}^{3/4} \overline{m}_{t-1} - \langle \boldsymbol{\theta}^*, \mathbf{x}_t^{\mathcal{H}} \rangle$$

$$\overset{(24)}{\leq} \frac{1}{c_{\mathrm{H}}}\left(\langle\widehat{\boldsymbol{\theta}}_{t-1}, \mathbf{x}_t^{\mathcal{H}}\rangle + \frac{\beta_{t-1}^{1/4}}{4c_0\overline{m}_{t-1}}\|\mathbf{x}_t^{\mathcal{H}}\|_{\overline{\mathbf{V}}_{t-1}^{-1}}^2\right) + c_0\beta_{t-1}^{3/4}\overline{m}_{t-1} - \langle\boldsymbol{\theta}^*, \mathbf{x}_t^{\mathcal{H}}\rangle$$

$$= \left(\frac{1}{c_{\mathrm{H}}}-1\right)\langle\widehat{\boldsymbol{\theta}}_{t-1}, \mathbf{x}_t^{\mathcal{H}}\rangle + \langle\widehat{\boldsymbol{\theta}}_{t-1} - \boldsymbol{\theta}^*, \mathbf{x}_t^{\mathcal{H}}\rangle + \frac{1}{c_{\mathrm{H}}}\frac{\beta_{t-1}^{1/4}}{4c_0\overline{m}_{t-1}}\|\mathbf{x}_t^{\mathcal{H}}\|_{\overline{\mathbf{V}}_{t-1}^{-1}}^2 + c_0\beta_{t-1}^{3/4}\overline{m}_{t-1}$$

$$\overset{(25)}{\leq} \underbrace{\frac{\log T}{\sqrt{T}}S}_{A_1(t)} + \underbrace{\frac{\log T}{\sqrt{T}}\sqrt{\overline{\beta}_{t-1}}\|\mathbf{x}_t^{\mathcal{H}}\|_{\overline{\mathbf{V}}_{t-1}^{-1}}}_{A_2(t)} + \underbrace{\langle\widehat{\boldsymbol{\theta}}_{t-1} - \boldsymbol{\theta}^*, \mathbf{x}_t^{\mathcal{H}}\rangle + \frac{1}{c_{\mathrm{H}}}\frac{\beta_{t-1}^{1/4}}{4c_0\overline{m}_{t-1}}\|\mathbf{x}_t^{\mathcal{H}}\|_{\overline{\mathbf{V}}_{t-1}^{-1}}^2 + c_0\beta_{t-1}^{3/4}\overline{m}_{t-1}}_{A_3(t)} \ .$$

Note that $r_t \leq 2LS$. Since $1/c_{\mathrm{H}} \leq 2$, it is not hard to see that $\sum_{t=1}^T \min\{2LS, L \cdot A_3(t)\}$ is

$$O(\kappa^{-1}L(L+R)d^{5/4}\sqrt{T}\log^{7/4}(T))$$

using the same technique as the proof of Theorem 5. It remains to bound $\sum_{t=1}^T \min\{2LS, L \cdot A_1(t)\}$ and $\sum_{t=1}^T \min\{2S, L \cdot A_2(t)\}$:

$$L\sum_{t=1}^T \min\{2S, A_1(t)\} \leq L\sum_{t=1}^T \min\{2S, \frac{\log T}{\sqrt{T}}2S\}$$

$$\overset{(\log T < \sqrt{T})}{\leq} L\sum_{t=1}^T \frac{\log T}{\sqrt{T}}2S$$

$$= O(L\sqrt{T}\log T)$$

$$L\sum_{t=1}^T \min\{2S, A_2(t)\} = L\sum_{t=1}^T \min\left\{2S, \frac{\log T}{\sqrt{T}}\sqrt{\overline{\beta}_{t-1}}\|\mathbf{x}_t^{\mathcal{H}}\|_{\overline{\mathbf{V}}_{t-1}^{-1}}\right\}$$

$$\leq L\sum_{t=1}^T \min\left\{2S, \sqrt{\overline{\beta}_{t-1}}\|\mathbf{x}_t^{\mathcal{H}}\|_{\overline{\mathbf{V}}_{t-1}^{-1}}\right\}$$

Using the same argument as the proof of Theorem 2 and Corollary 2,

$$\sum_{t=1}^T \min\{2S, L \cdot A_2(t)\} = O\left(\frac{L(L+R)}{\kappa}d\sqrt{T}\log^{3/2}(T)\right) \ .$$

Altogether, we notice that $\sum_{t=1}^T \min\{2S, L \cdot A_3(t)\}$ dominates the other terms. Notice that we have spent $\delta$ probability to control the event $\boldsymbol{\theta}^* \in C_t^{\mathrm{ONS}}$, another $\delta$ for controlling the deviation of $g_t$ which appears in the regret bound through $\beta_t$ as shown in the proof of Theorem 2, and another $\delta$ for ensuring the hashing guarantee. This sums to $3\delta$ and concludes the proof. $\qquad\square$

## L   Proof of Lemma 1

*Proof.* We first recall that for L1 sampling:

$$p_i = \frac{|q_i|}{\|\mathbf{q}\|_1}$$

Note that:

$$|G_k| = \left|\frac{q_{i_k}a_{i_k}}{p_{i_k}}\right|$$

$$= \frac{|q_{i_k}a_{i_k}|}{|q_{i_k}|/\|\mathbf{q}\|_1}$$

$$= \|\mathbf{q}\|_1 |a_{i_k}|$$

$$\leq \|\mathbf{q}\|_1 \|\mathbf{a}\|_{\max} =: M$$

This means that $G_k$ is a bounded random variable. Therefore, we can use Hoeffding's inequality to get a high-probability bound:

$$X_i = G_i - \mathbf{q}^\top\mathbf{a}$$

$$\mathbb{P}\left(\left|\frac{1}{m}\sum_{i=1}^{m}X_i\right|\geq\epsilon\right)\leq 2\exp\left(-\frac{m\epsilon^2}{2M^2}\right)$$

$$\leq 2\exp\left(-\frac{m\epsilon^2}{2\left\|\mathbf{q}\right\|_1^2\left\|\mathbf{a}\right\|_{\max}^2}\right)$$

$\square$

## M Comparison between L1 and L2 sampling

We first show the known high probability error bound of L2, which has a polynomially decaying probability of failure.

**Lemma 7.** *[22, Lemma 3.4] Define $G_k$ as in* (11) *with* $\mathbf{p}=\mathbf{p}^{(\text{L2})}$. *Then, given a target error $\epsilon>0$,*

$$\mathbb{P}\left(\left|\frac{1}{m}\sum_{k=1}^{m}G_k-\mathbf{q}^\top\mathbf{a}\right|\geq\epsilon\right)\leq\frac{||\mathbf{q}||_2^2||\mathbf{a}||_2^2}{m\epsilon^2}.\tag{26}$$

*Proof.* This proof is a streamlined version of the proof of Lemma 3.4 from [22]. Note that

$$\mathrm{Var}G_k\leq\left\|\mathbf{q}\right\|^2\left\|\mathbf{a}\right\|^2.$$

Define

$$X_m:=\sum_{i=1}^{m}\left(G_i-\mathbf{q}^\top\mathbf{a}\right)$$

Now, since $\mathbf{q}^\top\mathbf{a}$ is deterministic,

$$\mathrm{Var}X_m=\sum_{i=1}^{m}\mathrm{Var}G_i\leq m\left\|\mathbf{q}\right\|^2\left\|\mathbf{a}\right\|^2$$

and,

$$\mathbb{E}X_m=0$$

Let us apply Chebyshev's inequality to $X_m$:

$$\mathbb{P}\left(\left|X_m-\mathbb{E}X_m\right|\geq\alpha\right)\leq\frac{\mathrm{Var}X_m}{\alpha^2}$$

$$\Rightarrow\mathbb{P}\left(\left|X_m\right|\geq m\epsilon\right)\leq\frac{m\left\|\mathbf{q}\right\|^2\left\|\mathbf{a}\right\|^2}{m^2\epsilon^2}$$

$$\Rightarrow\mathbb{P}\left(\left|\frac{1}{m}\sum_{i=1}^{m}G_i-\mathbf{q}^\top\mathbf{a}\right|\geq\epsilon\right)\leq\frac{\left\|\mathbf{q}\right\|^2\left\|\mathbf{a}\right\|^2}{m\epsilon^2}$$

$$\Rightarrow\mathbb{P}\left(\left|\frac{1}{m}\sum_{i=1}^{m}G_i-\mathbf{q}^\top\mathbf{a}\right|\geq\epsilon'\left\|\mathbf{q}\right\|\left\|\mathbf{a}\right\|\right)\leq\frac{\left\|\mathbf{q}\right\|^2\left\|\mathbf{a}\right\|^2}{m\epsilon'^2\left\|\mathbf{q}\right\|^2\left\|\mathbf{a}\right\|^2},\text{ where }\epsilon=\epsilon'\left\|\mathbf{q}\right\|\left\|\mathbf{a}\right\|$$

$$\Rightarrow\mathbb{P}\left(\left|\tilde{\mathbf{q}}^\top\mathbf{a}-\mathbf{q}^\top\mathbf{a}\right|\geq\epsilon'\left\|\mathbf{q}\right\|\left\|\mathbf{a}\right\|\right)\leq\frac{1}{m\epsilon'^2}$$

$$\Rightarrow\mathbb{P}\left(\left|\tilde{\mathbf{q}}^\top\mathbf{a}-\mathbf{q}^\top\mathbf{a}\right|\geq\epsilon'\left\|\mathbf{q}\right\|\left\|\mathbf{a}\right\|\right)\leq\frac{1}{c},\text{ where }m=c/\epsilon'^2$$

$$\Rightarrow\mathbb{P}\left(\left|\tilde{\mathbf{q}}^\top\mathbf{a}-\mathbf{q}^\top\mathbf{a}\right|\geq\epsilon'\left\|\mathbf{q}\right\|\left\|\mathbf{a}\right\|\right)\geq 1-\frac{1}{c}$$

$\square$

To compare the concentration of measure of L1 and L2, we look at so-called "sample complexity" of L1 and L2. Let $\delta'$ be the target failure probability (set the RHS of (13) to $\delta'$). Then, the sample complexity of L2 and L1 is $\frac{||\mathbf{q}||_2^2||\mathbf{a}||_2^2}{\delta'\epsilon^2}$ and $\log(2/\delta')\frac{2||\mathbf{q}||_1^2||a||_{\max}^2}{\epsilon^2}$, respectively. Note L2 has a smaller scaling with $\mathbf{q}$ since $||\mathbf{q}||_2\leq||\mathbf{q}||_1$, but L1 has a better scaling with $\mathbf{a}$ since $||\mathbf{a}||_{\max}\leq||\mathbf{a}||_2$. More importantly, the concentration bound of L2 decays polynomially with $m$ whereas that of L1 decays exponentially. However, it is unclear whether or not the polynomial tail of L2 is just an artifact of analysis. In fact, we find that L2 has an exponential tail bound, but its scaling with $\mathbf{q}$ can be very bad.

While we state the result below in Lemma 8, the key here is that the magnitude of $G_k$ induced by L2 is at most $||\mathbf{q}||_2^2 \max_i |a_i|/|q_i|$ while that induced by L1 is at most $||\mathbf{q}||_1 ||\mathbf{a}||_{\max}$. As $q_i$ goes to zero, the support of the L2-based $G_k$ can be arbitrarily large. Unless one knows that $q_i$ is sufficiently bounded away from 0 (which is false in QGLOC), L2-based $G_k$ raises a concern of having a "thick" tail.[9]

**Lemma 8.** *Use* $\mathbf{p} = \mathbf{p}^{(L2)}$. *Then,*

$$\mathbb{P}\left(\left|\frac{1}{m}\sum_{i=1}^m X_i\right| \geq \epsilon\right) \leq 2\exp\left(-\frac{m\epsilon^2}{2||\mathbf{q}||_2^4 \max_i |a_i/q_i|^2}\right)$$

*Proof.* From the results in Lemma 3.4 of [22] (restated as Lemma 7 in our paper), it seems that L2 sampling has polynomial tails, which is considered as heavy-tailed. Here we show that the tail of L2 sampling does not decay polynomially but exponentially by using Hoeffding's inequality instead of Chebychev's inequality. However, the scale that controls the tail thickness is quite bad, and the tail can be arbitrarily thick regardless of the norm or variance of $\mathbf{q}$.

We first recall that for L2 sampling:

$$p_i = \frac{q_i^2}{||\mathbf{q}||_2^2}$$

Let us first show an upper bound for $|G_k|$:

$$|G_k| = \left|\frac{q_{i_k} a_{i_k}}{p_{i_k}}\right|$$

$$= \frac{|q_{i_k} a_{i_k}|}{q_{i_k}^2 / ||\mathbf{q}||_2^2}$$

$$= ||\mathbf{q}||_2^2 \left|\frac{a_{i_k}}{q_{i_k}}\right|$$

$$\leq ||\mathbf{q}||_2^2 \max_i \left|\frac{a_i}{q_i}\right| =: M'$$

Recall that $\mathbb{E}G_k = 0$ and from above, $|G_k| \leq M'$. We can now apply Hoeffding's inequality:

$$X_i = G_i - \mathbf{q}^\top \mathbf{a}$$

$$\mathbb{P}\left(\left|\frac{1}{m}\sum_{i=1}^m X_i\right| \geq \epsilon\right) \leq 2\exp\left(-\frac{m\epsilon^2}{2M'^2}\right)$$

$$\leq 2\exp\left(-\frac{m\epsilon^2}{2||\mathbf{q}||_2^4 \max_i \left|\frac{a_i}{q_i}\right|^2}\right)$$

This is an exponential tail bound compared to the polynomial tail bound in Lemma 3.4 in [22]. We do note that the term $\max_i \left|\frac{a_i}{q_i}\right|$ could be very bad if $\min_i q_i$ is small and the corresponding $a_i$ is non-zero. $\qquad\square$

The second comparison is on the variance of L1 and L2. The variance of $G_k$ based on $\mathbf{p}^{(L2)}$ can be shown to be $||\mathbf{q}||_2^2 ||\mathbf{a}||_2^2 - (\mathbf{q}^\top \mathbf{a})^2$. With L1, the variance of $G_k$ is now $||\mathbf{q}||_1 (\sum_{i=1}^d |q_i| a_i^2) - (\mathbf{q}^\top \mathbf{a})^2$ whose first term can be upper-bounded by $||\mathbf{q}||_1 ||\mathbf{q}||_2 ||\mathbf{a}||_4^2$ using Cauchy-Schwartz. This means that the variance induced by L1 scales larger with $\mathbf{q}$ than by L2 since $||\mathbf{q}||_2 \leq ||\mathbf{q}||_1$ and scales smaller with $\mathbf{a}$ since $||\mathbf{a}||_4^2 \leq ||\mathbf{a}||_2^2$. Thus, neither is an absolute winner. However, if the vectors being inner-producted are normally distributed, then L1 has a smaller variance than L2 in most cases, for large enough $d$ as we show in the lemma below. As mentioned in the main text, our projection vectors are truly normally distributed.

**Lemma 9.** *Suppose that* $\mathbf{q} \sim \mathcal{N}(0, \mathbf{I}_d)$ *and* $\mathbf{a} \sim \mathcal{N}(0, \mathbf{I}_d)$ *and that* $\mathbf{q}$ *and* $\mathbf{a}$ *are independent of each other. Then* $\mathbb{E}\left( \|\mathbf{q}\|_1 \sum_{i=1}^d |q_i| \, a_i^2 \right) \leq \mathbb{E}\left( \|\mathbf{q}\|_2^2 \|\mathbf{a}\|_2^2 \right)$ *for large enough* $d$.

*Proof.* First note that if $\mathbf{x} \sim \mathcal{N}(0, I_d)$, then:

$$\mathbb{E}\|\mathbf{x}\|_2^2 = d .$$

Then, the RHS is

$$\mathbb{E}\|\mathbf{q}\|_2^2 \|\mathbf{a}\|_2^2 = \mathbb{E}\|\mathbf{q}\|_2^2 \, \mathbb{E}\|\mathbf{a}\|_2^2$$
$$= d^2 .$$

For the LHS:

$$\mathbb{E}\left( \|\mathbf{q}\|_1 \sum_{i=1}^d |q_i| \, a_i^2 \right) = \mathbb{E}_{\mathbf{q}}\left( \|\mathbf{q}\|_1 \sum_{i=1}^d |q_i| \, \mathbb{E}_{\mathbf{a}} a_i^2 \right)$$
$$= \mathbb{E}_{\mathbf{q}}\left( \|\mathbf{q}\|_1 \sum_{i=1}^d |q_i| \right)$$
$$= \mathbb{E}_{\mathbf{q}} \|\mathbf{q}\|_1^2$$
$$= \mathbb{E}_{\mathbf{q}}\left( \sum_{i=1}^d |q_i|^2 + 2 \sum_{i,j=1,i\neq j}^d |q_i| \, |q_j| \right)$$
$$= \mathbb{E}_{\mathbf{q}} \sum_{i=1}^d |q_i|^2 + 2 \sum_{i,j=1,i\neq j}^d \mathbb{E}_{\mathbf{q}} |q_i| \, |q_j|$$
$$= d + 2 \frac{2}{\pi} \frac{d(d-1)}{2}$$
$$= d + \frac{2}{\pi} d(d-1)$$
$$= \frac{2}{\pi} d^2 + \frac{\pi-2}{\pi} d .$$

where we used the fact that $\sum_{i=1}^d |q_i|^2$ follows a Chi-squared distribution with $d$ degrees of freedom and that $|q_i|$ follows a half-normal distribution, whose mean is $\sqrt{2/\pi}$.

Since $2/\pi < 1$, for large $d$, $\frac{2}{\pi} d^2 + \frac{\pi-2}{\pi} d < d^2$. $\qquad \square$

# N   Hashing Implementation

We implement hashing for QGLOC and GLOC-TS based on python package OptimalLSH[10]. In practice, building a series of hashings with varying parameter shown in Section J can be burdensome. Departing from the theory, we use so-called multi-probe technique [37] that achieves a similar effect by probing nearby buckets. That is, upon determining a hash key and retrieving its corresponding bucket, one can also look at other hash keys that are different on only one component. We use $k = 12$ keys and $U = 24$ tables.