[Reviews · NeurIPS 2017]

Reviewer 1



The paper identifies two problems with existing Generalized Linear Model Bandit algorithms: their per-step computational complexity is linear in t (time-steps) and N (number of arms). The first problem is solved by the proposed GLOC algorithm (first that achieves constant-in-t runtime and O(d T^½ polylog(T) ) regret) and the second by a faster approximate variant (QGLOC) based on hashing. Minor contributions are a (claimed non-trivial) generalization of linear online-to-confidence bounds to GLM, an exploration-exploitation tradeoff parameter for QGLOC that finally justifies a common heuristic used by practitioners and a novel simpler hashing method for computing QGLOC solutions. I think the paper is interesting and proposes a novel idea, but the presentation is sometimes confused and makes it hard to evaluate the impact of the contribution w.r.t. the existing literature. the authors claim that all existing GLMB algorithms have a per-step complexity of t. This is clearly not the case in the linear setting when \mu is the identity, and closed forms for \hat{theta}_t allow the solution to be computed incrementally. This cannot be done in general in the GLM where \hat{\theta}_t is the solution of a convex optimization problem that requires O(t) time to be created and solved. Indeed, GLOC itself uses ONS to estimate \theta, and then obtain a closed form solution for \hat{\theta}, sidestepping the O(t) convex problem. The authors should improve the presentation and comparison with existing methods, highlighting when and where this dependence on t comes from and why other methods (e.g., why hot restarting the convex solver with \hat{\theta}_{t-1}) cannot provide the same guarantees, or other examples in which closed forms can be computed. The authors should expand the discussion on line 208 to justify the non-triviality: what original technical results were necessary to extend Abbasi-Yadkori [2] to GLMs? While the road to solving the O(t) problem is somewhat clear, the approach to reduce the O(N) complexity is not well justified. In particular (as i understood): GLOC-TS is hash-amenable but has d^3/2 regret (when solved exactly) QGLOC is hash-amenable and has d^5/4 regret, but only when solved exactly GLOC-TS has no know regret bound when solved inexactly (using hashing) QGLOC has a fixed-budget regret guarantee (presented only in the appendix, d^5/4 or d^3/2?) that holds when solved inexactly (using hashing), but reintroduces a O(t) space complexity (no O(t) time complexity?) This raises a number of questions: does GLOC-TS have a regret bound when solved inexactly? Does any of these algorithms (GLOC, QGLOC, GLOC-TS) guarantee O(d^{1,5/4,3/2}\sqrt{T}) regret while using sublinear space/time per step BOTH in t and N at the same time? The presentation should also be improved (e.g., L1-sampling requires choosing m, but QGLOC's theorem makes no reference to how to choose m) The experiments report regret vs. t (time), and I take t here is the number of iterations of the algorithm. In this metric, UCB-GLM (as expected) outperforms the proposed method. It is important to report metrics (runtime) that were the reason for the introduction of GLOC, namely per-step time and space complexity, as t and N change. Minor things: L230 "certain form" is a vague term L119 S must be known in advance, it is very hard to estimate, and has a linear impact on the regret. How hard it is to modify the algorithm to adapt to unknown S? L109+L127 The loss fed to the online learner is \ell(x,y) = -yz +m(z) and m(z) is \kappa strongly convex under Ass. 1. For strongly convex functions, ONS is not necessary and simple gradient descent achieves logarithmic regret. Under appropriate conditions on \kappa and y, the problem might become simpler. L125 Ass. 1 is common in the literature, but no intuition for a value of \kappa is provided. Since the regret scales linearly with \kappa, it would be useful to show simple examples of known \kappa e.g. in the probit model and 0/1 rewards used in the experiments. It would also be useful to show that it does not satisfy the previous remark. L44 3 is not included in the "existing algorithm". Typo, or TS does not suffer from both scalability issues? L158 S should be an input of Alg. 1 as well as Alg. 2 AFTER REBUTTAL I have read the author rebuttal, and I confirm my accept. I believe the paper is interesting, but needs to make a significant effort to clarify the message on what it does and what it does not achieve.

Reviewer 2



--- Paper --- The paper focuses on computationally efficient versions of Generalized Linear Bandits. It builds upon online learning to get rid of complexity increasing with time, it proposes two ways to deal with the number of arms, and it discusses its own flavor of approximate inner product. The price to pay is about the regret of the proposed approaches, which ranges from the same as state of the art to a regret with an additional $d$ factor depending on the algorithm and on the setting (infinite of finite number of arms). As most applications involve a finite set of arms, we may argue that the proposed approches suffer at least a $\sqrt(d)$ additional term. --- Review --- The aim of the paper deserves interest and the proposed meta-algorithm path the way to several applications. Its a pleasure to see computational limitations studied in any corner. Compared to previous attempts concerning Generalized Linear Bandits [41], the proposed approach is generic: Theorems 1 and 2 apply to any couple (Generalized Linear Model, Online Learning algorithm). The only missing result could be a problem-dependent bound of the regret, which anyway loses its interest when the number of arms is big. --- Typos --- L50: a specific 0/1 reward[s] with the logistic L319: where we [we] scale down

Reviewer 3



Overview: The paper proposes new, scalable algorithms for generalized linear bandits. The existing GLB algorithms have scalability issues (i) under a large time horizon, and (ii) under a large number of arms: the per-time-step space and time complexity of existing GLB algorithms grow at least linearly with time t, and existing algorithms have linear time complexities in the number of arms N. To deal with the first issue, the paper proposes to use any online learning algorithm as a black box, turn the outputs of the online learner into a confidence set of the true parameter, and finally apply the principle of optimism in the face of uncertainty. The paper provides a regret bound for the proposed algorithm. To deal with the second problem, the paper proposes to use hashing as a solution. The paper gives two GLB algorithms that are hash-amenable. The paper also proposes a sampling-based method to reduce the overhead of hashing. Significance: The paper contributes to making GLB algorithms more scalable and more practical for real-world applications that require fast decisions. Originality: The paper mostly extends existing tools and techniques to GLB problems. Quality: The paper provides mathematical proofs for all its claims. The paper also includes a section of computational studies for its proposed algorithms. The part on GLOC looks solid, while the part on hash-amenable GLBs seems insufficient. The authors may want to compare QGLOC-Hash and GLOC-TS-Hash with, say, GLOC and UCB-GLM to show how performance deteriorates with the use of hashing. Clarity: While the content of this paper may be of great practical interests, this paper is not very well-written. The authors should clearly define the notations and symbols before including them in the theorems. For example, the theta_t_hat in Theorem 1 is not defined anywhere in the text. Also, in Theorem 2, it is unclear what beta_t_bar does. There is also a typo in Theorem 3, etc. Further, the authors may want to give a formal definition of hash-amenability in section 4 and explains how hashing works with the rest of the bandit algorithm. Besides stating all the theorems, the authors may want to better explain their approaches in words. For example, when constructing the confidence set, why do you choose the center of the ellipsoid to be the ridge regression estimator on the natural parameters predicted by the online learner? The authors may want to give more justifications (in words) for their approaches so that the reader can better understand their algorithms.